# Boosting Out-of-distribution Detection with Typical Features

**Yao Zhu**[1,2] *   **Yuefeng Chen**[2]   **Chuanlong Xie**[3]   **Xiaodan Li**[2]   **Rong Zhang**[2]
**Hui Xue**[2]   **Xiang Tian**[1,5] †   **Bolun Zheng**[4,5]   **Yaowu Chen**[1,6]

[1]Zhejiang University, [2]Alibaba Group, [3]Beijing Normal University, [4]Hangzhou Dianzi University
[5]Zhejiang Provincial Key Laboratory for Network Multimedia Technologies
[6] Zhejiang University Embedded System Engineering Research Center, Ministry of Education of China

## Abstract

Out-of-distribution (OOD) detection is a critical task for ensuring the reliability and safety of deep neural networks in real-world scenarios. Different from most previous OOD detection methods that focus on designing OOD scores or introducing diverse outlier examples to retrain the model, we delve into the obstacle factors in OOD detection from the perspective of typicality and regard the feature's high-probability region of the deep model as the feature's typical set. We propose to rectify the feature into its typical set and calculate the OOD score with the typical features to achieve reliable uncertainty estimation. The feature rectification can be conducted as a plug-and-play module with various OOD scores. We evaluate the superiority of our method on both the commonly used benchmark (CIFAR) and the more challenging high-resolution benchmark with large label space (ImageNet). Notably, our approach outperforms state-of-the-art methods by up to 5.11% in the average FPR95 on the ImageNet benchmark [3].

## 1   Introduction

Deep neural networks have been widely applied in various fields. Apart from the success of deep models, predictive uncertainty is essential in safety-critical real-world scenarios such as autonomous driving [1, 2], medical [3], financial [4], etc. When encountering some examples that the deep model has not been exposed to during training, we hope the model raises an alert and hands them over to humans for safe handling. Such a challenge is usually referred to as out-of-distribution (OOD) detection and has gained significant research attention recently [5–8].

Most of the existing research [5–7, 9–12] worked on designing suitable OOD scores for the pretrained neural network, hoping to assign higher scores to the in-distribution (ID) examples and lower scores to the out-of-distribution (OOD) examples. However, these methods overlook the obstacle factors in OOD detection caused by the model's internal mechanisms. In this paper, we rethink the OOD detection from a perspective of feature typicality. We observed that the distribution of the deep features of the training dataset on different channels is approximately consistent with the Gaussian distribution (See examples in Appendix J). Accordingly, we divide these features into typical features (fall in the high-probability region) and extreme features (fall in the low-probability region). Extreme features rarely appear in training and attract less attention from the classifier than the typical features. We hypothesize the classifier can model the typical features better than the extreme features, and the extreme features may lead to ambiguity and imprecise uncertainty estimation. Given the potential

---

*Yao Zhu is with the Zhejiang University, Hangzhou, China, 310013. (E-mail: ee_zhuy@zju.edu.cn).

†Corresponding authors: Xiang Tian, Bolun Zheng. (E-mail: tianx@zju.edu.cn, blzheng@hdu.edu.cn)

[3]The code will be available at this https URL

negative impact, properly dealing with these extreme features is a key to improving the performance of OOD detection.

In this paper, we propose to rectify the features into their typical set and then calculate the OOD score with these typical features. In this way, the model conservatively utilizes the typical features to make decisions and alleviates the damage caused by extreme features, which can be beneficial to the OOD scores derived from the pre-trained classifier [5–7, 12]. Then the problem is how to estimate the feature's typical set on different channels since this requires a sufficient number of in-distribution examples and is time-consuming. Luckily, the commonly used operation Batch Normalization can shed light on a shortcut to selecting the feature's typical set and we name our approach **B**atch Normalization **A**ssisted **T**ypical **S**et Estimation (**BATS**). The Batch Normalization layer endeavors to normalize the features of the training dataset to Gaussian distributions, which can be used to estimate the typical set for ID features. Typical features are more common in training, while extreme features are rare, which leads to difficulties for the model to estimate extreme features well. We truncate the deep features with the guidance of the Batch Normalization, rectifying the extreme features to the boundary values of typical sets. We illustrate the distribution of the OOD scores for ID (ImageNet) and OOD (four different datasets) examples in Fig. 1. Rectifying the features into the typical set with our **BATS** contributes to improving the separability between ID and OOD examples.

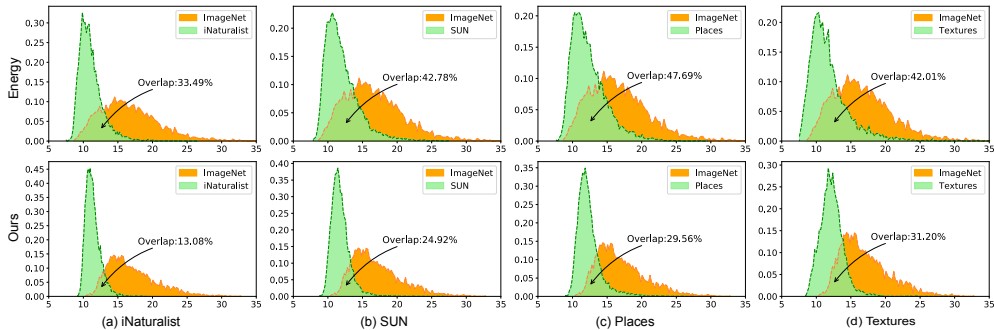

Figure 1: The distribution of the scores for ID (ImageNet) and OOD examples on ResNet-50. We use the energy score [5] as the OOD score. "Energy" means calculating the OOD score with the original features. "Ours" means calculating the OOD score with the typical features.

Theoretically, we analyze the benefit of **BATS** and the bias-variance trade-off influenced by the strength of the hyperparameter. A proper strength of **BATS** contributes to improving the estimation accuracy of the reject region. Empirically, we perform extensive evaluations and establish superior performance on both the large-scale ImageNet benchmark and the commonly used CIFAR benchmarks. **BATS** outperforms the previous best method by a large margin, with up to a 5.11% reduction in the false positive rate (FPR95) and a 1.43% improvement in AUROC. Moreover, **BATS** can also slightly improve the test accuracy and robustness of the pre-trained models. The main contributions of our paper are summarized as follows:

- We provide novel insights into OOD detection from the perspective of typicality and propose to rectify the features into the typical set. We design a concise and effective approach to select the feature's typical set named **B**atch Normalization **A**ssisted **T**ypical **S**et Estimation (**BATS**).

- We provide theoretical analysis and empirical ablation on the benefit of **BATS** from the perspective of bias-variance trade-off to improve the understanding of our approach.

- Extensive experiments show that **BATS** establishes a state-of-the-art performance among post-hoc methods on a suite of OOD detection benchmarks. Moreover, **BATS** can boost the performance of various existing OOD scores with typical features.

## 2 Related work

The literature related to OOD detection can be broadly grouped into the following themes: post-hoc detection methods [5–8, 11–13], confidence enhancement methods [9, 10, 14–19], and density-based methods [20–25]. Post-hoc detection methods focus on improving the OOD uncertainty estimation

by utilizing the pre-trained classifiers rather than retraining a model, which is beneficial for adopting OOD detection in real-world scenarios and large-scale settings. MSP [6] observes that the maximum softmax probability of ID examples can be higher than that of the OOD examples and provide a simple baseline for OOD detection. ODIN [7] introduces a sufficiently large temperature factor and input perturbation to separate the ID and OOD examples. Liu et al. [5] analyze the limitations of softmax function in OOD detection and propose to use energy score as an indicator. The examples with high energy are considered OOD examples, and vice versa. ReAct [8] hypothesizes that the OOD examples can trigger the abnormal activation of the model and propose to clamp the activation value larger than the threshold to improve the detection performance. GradNorm [12] shows that the gradients of the categorical cross-entropy loss can be an effective test statistic for OOD detection. Different from these methods, our **BATS** proposes to calculate the OOD scores with the typical features, which benefits the estimation of the reject region and can improve the detection performance.

## 3 Preliminaries

### 3.1 Out-of-distribution detection

In this section, we provide a summary of the out-of-distribution detection from the perspective of hypothesis testing [26–30]. We consider a classification problem with $K$ classes and denote the labels as $\mathcal{Y} = \{1, 2, \ldots, K\}$. Let $\mathcal{X}$ be the input space. Suppose that the in-distribution data $\mathcal{D}_{in} = \{(x_i, y_i)\}_{i=1}^n$ is drawn from a joint distribution $P_{X,Y}$ defined over $\mathcal{X} \times \mathcal{Y}$. We denote the marginal distribution of $P_{X,Y}$ for the input variable $X$ by $P_0$. Given a test input $\mathbf{x} \in \mathcal{X}$, the problem of out-of-distribution detection can be formulated as a single-sample hypothesis testing task:

$$\mathcal{H}_0 : \mathbf{x} \sim P_0, \quad \text{vs.} \quad \mathcal{H}_1 : \mathbf{x} \nsim P_0. \tag{1}$$

Here the null hypothesis $\mathcal{H}_0$ implies that the test input $\mathbf{x}$ is an in-distribution sample. The goal of OOD detection here is to design criteria based on $\mathcal{D}_{in}$ to determine whether $\mathcal{H}_0$ should be rejected. OOD detection tasks need to determine a reject region $\mathcal{R}$ such that for any test input $\mathbf{x} \in \mathcal{X}$, the null hypothesis is rejected if $x \in \mathcal{R}$. Generally, the reject region $\mathcal{R}$ is formulated by a test statistic and a threshold. Let $f : \mathcal{X} \mapsto \mathbb{R}^K$ be a model pre-trained from $\mathcal{D}_{in}$, which is used to predict the class label of an input sample. One can use the model $f$ or a part of $f$ (e.g., feature extractor) to construct a test statistic $T(\mathbf{x}; f)$, where $\mathbf{x}$ is the test input. Then the reject region can be written as $\mathcal{R} = \{\mathbf{x} : T(\mathbf{x}; f) \leq \gamma\}$, where $\gamma$ is the threshold.

### 3.2 OOD detection with energy score

For a classifier $f$ and a data point $(\mathbf{x}, y)$, we use $f(\mathbf{x})[k]$ to represent the $k^{th}$ output of the last layer. With reference to [5, 31, 32], the marginal density $p(\boldsymbol{x})$ of the classifier can be expressed as: $p(x) = \frac{\exp(-E(x))}{Z} = \frac{\sum_{k=1}^K \exp(f(\mathbf{x})[k])}{Z}$, where $Z$ is the normalizing factor and is independent to $\mathbf{x}$. $E(x)$ represents the energy of $x$ and is modeled by neural network as $E(\mathbf{x}) = -\log \sum_{k=1}^K \exp(f(\mathbf{x})[k])$. See Appendix C for details. Considering that $Z$ is a constant and is independent to $\mathbf{x}$, Liu et al. [5] propose an energy score that uses the opposite of the energy $E(\mathbf{x})$ as a test statistic to detect OOD examples. A higher energy score means a higher marginal density $p(\boldsymbol{x})$.

## 4 Methods

In this paper, we delve into the obstacle factor for the post-hoc OOD detection from the perspective of typicality, which aims to boost the performance of the existing OOD scores and is orthogonal to the methods of designing different OOD scores. Given that the energy score [5] is provably aligned with the density of inputs and performs well, we mainly use the energy score as the OOD score. (See Appendix I for other OOD scores).

### 4.1 Motivation

For a classifier trained on the ID data $f = f_{w,b} \circ g$ where $g$ is a feature extractor mapping input $\mathbf{x}$ to its deep feature $\mathbf{z}$. Let $d$-dimensional vector $\mathbf{z} = [z_1, ..., z_d]^\top = g(\mathbf{x})$ denote the deep features of $\mathbf{x}$

extracted by $g$, and $z_i$ indicate the $i$-th element of $\mathbf{z}$. $f_{\mathbf{w},\mathbf{b}}(\mathbf{z}) = \mathbf{w} \cdot \mathbf{z} + \mathbf{b}$ is a fully connected layer mapping the deep feature $\mathbf{z}$ to output logits. The energy can be expressed as:

$$E(\mathbf{x}) = -\log \sum_{k=1}^{K} \exp(f(\mathbf{x})[k]) = -\log \sum_{k=1}^{K} \exp((\mathbf{w} \cdot \mathbf{z} + \mathbf{b})[k]). \tag{2}$$

The test statistic can be expressed as $T(\mathbf{x}; f) := -E(\mathbf{x}) = \log \sum_{k=1}^{K} \exp((\mathbf{w} \cdot \mathbf{z} + \mathbf{b})[k])$, which depends on the extracted deep features and the mapping operation of the fully connected layer (FC). Assuming that the distribution of the deep features is consistent with the Gaussian distribution (see examples in Appendix J), there are high-probability regions and low-probability regions in deep features. We name the features that fall in high-probability regions as typical features, and the corresponding regions are called feature's typical sets. In contrast, we regard the features that fall in low-probability regions as extreme features. Extreme features are rarely exposed to the training process, which leads to difficulties for the classifier to model these features and unreliable estimations in the inference process. Reducing the influence of extreme features on test statistics can be a key to improving OOD detection performance.

## 4.2 Batch Normalization Assisted Typical Set Estimation

Instead of designing new OOD scores to detect the abnormality, we provide a novel insight into OOD detection from a perspective of typicality. We propose to rectify the features into the feature's typical set and then use these typical features to calculate the OOD score. Consider a commonly used layer structure in deep convolutional networks:

$$\mathbf{z}' \to \mathrm{BN}(\mathbf{z}'; \mu, \sigma) \to \mathrm{ReLU} \to \mathbf{z}, \tag{3}$$

where $\mathbf{z}'$ is the feature vector extracted from the convolutional layer of $g$. To identify the typical set of $\mathbf{z}'$ for each channel, we should apply its feature map to a sufficient number of ID examples and further calculate the empirical distribution of $\mathbf{z}'$ over the ID examples. If the number of features is large, the inference procedure is time-consuming. *Here we propose a simple and effective post hoc approach that leverages the information stored in $f$ to infer the typical set without estimating the distribution of $\mathbf{z}'$.* Suppose the pre-trained deep neural network uses batch normalization (BN). We denote the BN unit in $f$ as:

$$\mathrm{BN}(\mathbf{z}'; \mu, \sigma) = \sigma \frac{\mathbf{z}' - \mathbb{E}(\mathbf{z}')}{\mathrm{Std}(\mathbf{z}')} + \mu, \tag{4}$$

where $\mu, \sigma$ are two learnable parameters. After the pre-training, all the four parameters $\mu$, $\sigma$, $\mathbb{E}(\mathbf{z}')$, $\mathrm{Std}(\mathbf{z}')$ are known and stored in the weights of $f$.[4] The Batch Normalization normalizes features of the training dataset to a distribution with a mean of $\mu$ and standard deviation of $\sigma$, which means that the features fall in the interval $[\mu - \lambda * \sigma, \mu + \lambda * \sigma]$ appear more frequently in training than the features in the complement of this interval. The parameter $\lambda$ controls the range of the interval. Thus we use the information in the Batch Normalization to identify the in-distribution feature's typical set and rectify the features into the typical set before calculating the OOD score. The uncertainty estimated with the typical features can be more reliable.

In practice, we propose a truncated activation scheme to bound the output features of the BN unit. First, we introduce the truncated BN unit by:

$$\mathrm{TrBN}(\mathbf{z}'; \mu, \sigma, \lambda) = \begin{cases} \mu + \lambda\sigma, & \text{if} \quad \mathbf{z}' - \mu \geq \lambda\sigma; \\ \mathrm{BN}(\mathbf{z}'; \mu, \sigma), & \text{if} \quad -\lambda\sigma < \mathbf{z}' - \mu < \lambda\sigma; \\ \mu - \lambda\sigma, & \text{if} \quad \mathbf{z}' - \mu \leq -\lambda\sigma, \end{cases} \tag{5}$$

where $\lambda$ is a tuning parameter. We replace the BN unit in the layer structure (Eq.(3)) with the TrBN unit and write the rectified final features as $\bar{\mathbf{z}}$ and the new classifier as $\bar{f}$. Then test statistic with the energy score can be expressed as:

$$T(\mathbf{x}; \bar{f}) = \log \sum_{k=1}^{K} \exp((\mathbf{w} \cdot \bar{\mathbf{z}} + \mathbf{b})[k]) \tag{6}$$

---

[4]In general, $\mathbb{E}(\mathbf{z}')$ and $\mathrm{Std}(\mathbf{z}')$ are estimated on a mini-batch of the training data. Finally, the pre-trained model outputs moving average estimators at each iteration.

and take the reject region by $\mathcal{R} = \{\mathbf{x} : T(\mathbf{x}; \bar{f}) \leq \gamma\}$. We name our approach **B**atch Normalization **A**ssisted **T**ypical **S**et Estimation (BATS). In comparison to the standard BN, the outputs of TrBN are concentrated toward the feature's typical set of ID data. This makes an ID example less susceptible to being mistakenly detected as an OOD example and buffers the negative impact of the extreme features. Fig. 1 compares the distribution of the OOD scores from the original energy score and the energy score with typical features.

## 4.3 Theoretical analysis

The truncation threshold $\lambda$ is a key hyperparameter. Our method reduces the variance of $\mathbf{z}'$ and also introduces a bias term since it changes the distribution of $\mathrm{BN}(\mathbf{z}'; \mu, \sigma)$. The variance reduction means that our method is robust to the rare ID examples, while the introduced bias can lead to degradation of the model performance. In this section, we assume $\mathbf{z}'$ follows a normal distribution and analyze the bias-variance trade-off in our method.

### 4.3.1 Understanding the benefits of BATS from the perspective of variance reduction

The variance reduction happens at the BN step. The variance of $\mathrm{BN}(\mathbf{z}'; \mu, \sigma)$ is $\sigma^2$ since the distribution of the ID features $\mathbf{z}'$ is rescaled to $N(\mu, \sigma^2)$. While the TrBN unit truncates the extreme values and the variance of $\mathrm{TrBN}(\mathbf{z}'; \mu, \sigma, \lambda)$ becomes:

$$\sigma^2 C(\lambda) := \sigma^2 \Big( \mathrm{erf}(\frac{\lambda}{\sqrt{2}}) - \frac{\sqrt{2}}{\sqrt{\pi}} \lambda \exp(-\frac{\lambda^2}{2}) + \lambda^2 (1 - \mathrm{erf}(\frac{\lambda}{\sqrt{2}})) \Big), \qquad (7)$$

where $\mathrm{erf}(x) = (2/\sqrt{\pi}) \int_0^x \exp(-t^2) dt$ is the Gauss error function. The value of $C(\lambda)$ represents the degree of variance reduction. In Eq.(7), $C(0) = 0$, $dC(\lambda)/d\lambda > 0$, and $C(\lambda) \to 1$ as $\lambda \to +\infty$. Therefore, $C(\lambda)$ is a monotonically increasing function and $0 \leq C(\lambda) < 1$ for $0 \leq \lambda < +\infty$. In summary, the smaller $\lambda$, the smaller the variance. See Appendix D for the proof.

OOD detection is a single-sample hypothesis testing problem (in Eq.(1)), and the in-distribution $P_0$ is unknown. So the reject region is determined by the empirical distribution of the test statistic $T(\mathbf{x}; f)$ over the ID data. The extreme features increase the uncertainty and lead to more unusual values of $T(\mathbf{x}; f)$. This implies that the reject region may be underestimated due to the heavy tail property of $T(\mathbf{x}; f)$. Our **BATS** aids this problem by reducing the variance of the deep features, which contributes to constraining the uncertainty of $f$ and $T(\mathbf{x}; f)$ and improving the estimation accuracy of the reject region.

### 4.3.2 The bias introduced by BATS

**BATS** rectifies the features into the typical set, which reduces the variance of the deep features. However, this operation can also introduce a bias term, which can reflect the change in the distribution of the features. A large bias can damage the performance of the model. The distribution of the output feature $\mathbf{z} = \mathrm{ReLU}(\mathrm{BN}(\mathbf{z}'; \mu, \sigma, \lambda))$ is a one-side rectified normal distribution over $[0, +\infty)$. The expectation of $\mathbf{z}$ is:

$$\mathbb{E}(\mathbf{z}) = \mu + \sigma \Big( \frac{1}{\sqrt{2\pi}} \big( \exp(-\frac{\mu^2}{2\sigma^2}) \big) - \frac{\mu}{2\sigma} (1 + \mathrm{erf}(-\frac{\mu}{\sqrt{2}\sigma})) \Big). \qquad (8)$$

For our method, the distribution of the output feature $\bar{\mathbf{z}} = \mathrm{ReLU}(\mathrm{TrBN}(\mathbf{z}; \mu, \sigma, \lambda))$ is a two-sided rectified normal distribution over $[0, \mu + \lambda\sigma]$ and the expectation of $\bar{\mathbf{z}}$ is:

$$\mathbb{E}(\bar{\mathbf{z}}) = \mu + \sigma \Big( \frac{1}{\sqrt{2\pi}} \big( \exp(-\frac{\mu^2}{2\sigma^2}) - \exp(-\frac{\lambda^2}{2}) \big) - \frac{\mu}{2\sigma} (1 + \mathrm{erf}(-\frac{\mu}{\sqrt{2}\sigma})) + \frac{\lambda}{2} (1 - \mathrm{erf}(\frac{\lambda}{\sqrt{2}})) \Big). \quad (9)$$

Then the bias caused by the truncation is:

$$\mathbb{E}(\bar{\mathbf{z}}) - \mathbb{E}(\mathbf{z}) = \sigma \Big( - \exp(-\frac{\lambda^2}{2}) + \frac{\lambda}{2} (1 - \mathrm{erf}(\frac{\lambda}{\sqrt{2}})) \Big) = \big( \lambda - \lambda\Phi(\lambda) - \phi(\lambda) \big) \sigma, \qquad (10)$$

where $\phi(\cdot)$ and $\Phi(\cdot)$ are the probability density function (pdf) and cumulative distribution function (cdf) of the standard normal distribution. One can find that the bias term $\mathbb{E}(\bar{\mathbf{z}}) - \mathbb{E}(\mathbf{z})$ converges to zero as $\lambda \to \infty$. In other words, if $\lambda$ is large enough, the bias can be very small. Thus, there exists a bias-variance trade-off. See Appendix D for the proof.

A proper selection of $\lambda$ can improve the detection performance by significantly reducing the uncertainty (variance reduction) and slightly changing the distribution of the features (small bias). If $\lambda$ is large, $T(\mathbf{x}; \bar{f})$ uses more extreme features in both the ID and OOD data. As $\lambda$ tends to infinity, $T(\mathbf{x}; \bar{f})$ converges to $T(\mathbf{x}; f)$. Then **BATS** is the same to the original energy detection. If $\lambda$ is small, extreme features are removed from the test statistic $T(\mathbf{x}; \bar{f})$ while introducing a non-negligible bias. Because of the change in feature distribution, the detection method loses its power to identify OOD examples. Fig. 2 illustrates the distribution of OOD scores with different $\lambda$, which empirically verifies this trade-off.

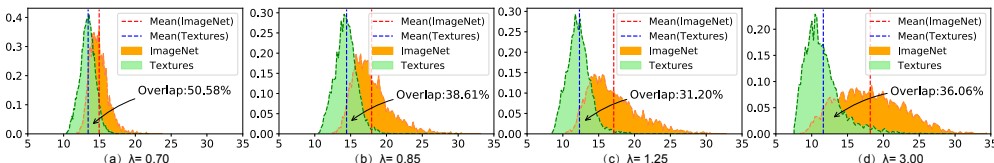

Figure 2: Bias-variance trade-off in BATS. We illustrate the OOD score for ID (ImageNet) and OOD (Textures) examples. Smaller $\lambda$ contributes to variance reduction which benefits the estimation of the reject region. But smaller $\lambda$ causes a larger bias, which can drastically alter the distribution of features and damage the performance of the model in distinguishing ID and OOD examples.

## 5 Experiments

In this section, we first introduce our experiment implementation. Then, we evaluate our methods both on the large-scale OOD detection benchmark [33] and the CIFAR benchmarks [6]. After that, the ablation studies compare the influence of applying rectification on different layers and show the influence of the hyperparameter. Moreover, our **BATS** can also slightly improve the test accuracy of the pre-trained models (in Appendix H). We consider the out-of-distribution detection as a **single-sample** hypothesis testing task and only test one sample at a time.

### 5.1 Implementation

**Dataset.** For evaluating the large-scale OOD detection performance, we use ImageNet-1k [33] as the in-distribution dataset and consider four out-of-distribution datasets, including (subsets of) the fine-grained dataset iNaturalist [34], the scene recognition datasets Places [35] and SUN [36], and the texture dataset Textures [37] with non-overlapping categories to ImageNet-1k.

As for the evaluation on CIFAR Benchmarks, we use the CIFAR-10 and CIFAR-100 [38] as the in-distribution datasets using the standard split with 50,000 training images and 10,000 test images. We consider four OOD datasets: SVHN [39], Tiny ImageNet [40], LSUN [41] and Textures [37].

**Baselines.** We consider different kinds of competitive OOD detection methods as baselines, including Maximum Softmax Probability (MSP) [6], ODIN [7], Energy [5], Mahalanobis [11], GradNorm [12] and ReAct [8]. MSP is a simple baseline for OOD detection and ReAct is a state-of-the-art method that achieves strong detection performance. All methods use the pre-trained networks post-hoc.

**Metrics. FPR95:** the false positive rate of OOD (negative) examples when the true positive rate of in-distribution (positive) examples is as high as 95%. Lower FPR95 indicates better OOD detection performance and vice versa. **AUROC:** the area under the receiver operating characteristic curve (ROC). Higher AUROC indicates better detection performance. See Appendix E for more details.

### 5.2 Evaluation on the large-scale OOD detection benchmark

We first evaluate our method on a large-scale OOD detection benchmark proposed by Huang and Li [33]. [33] revealed that OOD detection methods designed for the CIFAR benchmark might not effectively be adaptable for the ImageNet benchmark with a large semantic space. Recent literature [8, 12, 33] proposes to evaluate OOD detection performance on images that have higher resolution and contain more classes than the CIFAR benchmarks, which is more relevant to real-world applications.

In Tab. 1, we compare our method with the existing methods and show the OOD detection performance for each OOD test dataset and the average over the four datasets. We consider different

Table 1: OOD detection performance comparison on different architectures: ResNet-50 (RN50) [42], DenseNet-121 (DN121) [43] and MobileNet-V2 (MNet) [44]. We use the pre-trained models in PyTorch [45] trained on ImageNet. All methods are post hoc and can be directly used for pre-trained models. The best results are in Bold. The up arrow indicates that the higher the value, the better the performance, and vice versa.

| Model | Method | iNaturalist | | SUN | | Places | | Textures | | Average | |
|---|---|---|---|---|---|---|---|---|---|---|---|
| | | FPR95 ↓ | AUROC ↑ | FPR95 ↓ | AUROC ↑ | FPR95 ↓ | AUROC ↑ | FPR95 ↓ | AUROC ↑ | FPR95 ↓ | AUROC ↑ |
| RN50 | MSP[6] | 51.44 | 88.17 | 72.04 | 79.95 | 74.34 | 78.84 | 54.90 | 78.69 | 63.18 | 81.41 |
| | ODIN[7] | 41.07 | 91.32 | 64.63 | 84.71 | 68.36 | 81.95 | 50.55 | 85.77 | 56.15 | 85.94 |
| | Energy[5] | 46.65 | 91.32 | 61.96 | 84.88 | 67.97 | 82.21 | 56.06 | 84.88 | 58.16 | 85.82 |
| | Mahalanobis[11] | 97.00 | 52.65 | 98.50 | 42.41 | 98.40 | 41.79 | 55.80 | 85.01 | 87.43 | 55.47 |
| | GradNorm[12] | 23.73 | 93.97 | 42.81 | 87.26 | 55.62 | 81.85 | **38.15** | 87.73 | 40.08 | 87.70 |
| | ReAct[8] | 17.77 | 96.70 | 25.15 | 94.34 | 34.64 | **91.92** | 51.31 | 88.83 | 32.22 | 92.95 |
| | BATS(Ours) | **12.57** | **97.67** | **22.62** | **95.33** | **34.34** | 91.83 | 38.90 | **92.27** | **27.11** | **94.28** |
| DN121 | MSP[6] | 47.65 | 89.09 | 69.95 | 79.64 | 72.53 | 78.74 | 69.69 | 77.06 | 64.96 | 81.13 |
| | ODIN[7] | 30.72 | 93.66 | 57.90 | 86.11 | 63.16 | 83.54 | 53.51 | 83.88 | 51.32 | 86.80 |
| | Energy[5] | 33.16 | 93.81 | 53.79 | 86.70 | 61.01 | 83.83 | 55.42 | 84.06 | 50.85 | 87.10 |
| | Mahalanobis[11] | 97.36 | 42.24 | 96.21 | 41.28 | 97.32 | 47.27 | 62.78 | 56.53 | 88.42 | 46.83 |
| | GradNorm[12] | 22.88 | 94.40 | 43.12 | 87.55 | 55.80 | 82.00 | 47.58 | 85.16 | 42.35 | 87.28 |
| | ReAct[8] | 15.93 | 96.91 | 40.41 | 90.13 | 48.87 | 87.98 | 36.58 | 92.48 | 35.45 | 91.88 |
| | BATS(Ours) | **14.63** | **97.13** | **30.45** | **93.03** | **41.35** | **89.24** | **31.72** | **93.40** | **29.54** | **93.20** |
| MNet | MSP[6] | 63.09 | 85.71 | 79.67 | 76.01 | 81.47 | 75.51 | 75.12 | 76.49 | 74.84 | 78.43 |
| | ODIN[7] | 45.61 | 91.33 | 63.03 | 83.44 | 70.01 | 80.85 | 52.45 | 85.61 | 57.78 | 85.31 |
| | Energy[5] | 49.52 | 91.10 | 63.06 | 84.42 | 69.24 | 81.42 | 58.16 | 84.88 | 60.00 | 85.46 |
| | Mahalanobis[11] | 62.04 | 82.37 | 54.79 | 86.33 | 53.77 | 83.69 | 88.72 | 37.28 | 64.83 | 72.42 |
| | GradNorm[12] | 33.70 | 92.46 | 42.15 | 89.65 | 56.56 | 83.93 | **34.95** | **90.99** | 41.84 | 89.26 |
| | ReAct[8] | 37.08 | 93.41 | 53.13 | 86.04 | 54.15 | 83.31 | 42.45 | 89.42 | 46.70 | 88.05 |
| | BATS(Ours) | **31.56** | **94.33** | **41.68** | **90.21** | **52.43** | **86.26** | 38.69 | 90.76 | **41.09** | **90.39** |

architectures, including the widely used ResNet-50 [42], DenseNet-121 [43] and a lightweight model MobileNet-v2 [44]. Compared with the Energy Score [5], the difference in our approach is rectifying the features that deviate from the feature's typical set. Our method outperforms the Energy Score on ResNet-50 by 31.05% in FPR95 and 8.46% in AUROC. Furthermore, our method reduces FPR95 by 5.11% and improves AUROC by 1.33% compared to the state-of-the-art method [8] on ResNet-50. Here the models are pre-trained in a standard manner. We also show that **BATS** can boost the OOD detection when using the adversarially pre-trained classifiers in Appendix F.

Simultaneously, we observe that existing methods have different performances on different architectures. Specifically, the performance of the GradNorm [12] in FPR95 is 7.86% worse than that of ReAct [8] on the ResNet-50, but surpasses ReAct on MobileNet-V2 by 4.86%. Our method achieves the best performance on different architectures. Appendix K shows that our method also outperforms the existing methods when choosing the natural adversarial examples [46] as OOD examples.

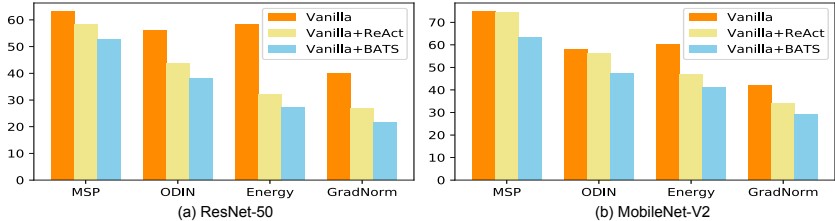

Figure 3: The FPR95 for different methods on ImageNet (lower is better) on ResNet-50 and MobileNet-V2. We illustrate the average performance on four OOD datasets. "Vanilla" means the original method and "Vanilla+BATS" means applying our **BATS** on the method.

Our experiments mainly use the energy score as the test statistic. In Fig. 3, we show that **BATS** is also compatible with various OOD scores and **BATS** can boost the performance of various OOD scores. Applying our **BATS** on GradNorm [12] (a gradient-based OOD score) can even achieve better performance than "Energy+BATS" but this method needs to derive the gradients of the model, which costs more than "Energy+BATS." See Appendix I for detailed performance.

Table 2: OOD detection performance on CIFAR-10 and CIFAR-100 [38]. All methods are post hoc and can be directly used for pre-trained models. The best results are in Bold.

| Dataset | Method | SVHN | | Tiny-Imagenet | | LSUN_resize | | Texture | | Average | |
|---|---|---|---|---|---|---|---|---|---|---|---|
| | | FPR95 ↓ | AUROC ↑ | FPR95 ↓ | AUROC ↑ | FPR95 ↓ | AUROC ↑ | FPR95 ↓ | AUROC ↑ | FPR95 ↓ | AUROC ↑ |
| CIFAR10 RN18 | MSP[6] | 59.60 | 91.29 | 50.01 | 93.02 | 52.15 | 92.73 | 66.63 | 88.50 | 57.10 | 91.39 |
| | ODIN[7] | 59.71 | 88.52 | **10.95** | **98.08** | **9.24** | **98.25** | 52.06 | 89.16 | 32.99 | 93.50 |
| | Energy[5] | 54.03 | 91.32 | 15.18 | 97.28 | 23.53 | 96.14 | 55.30 | 89.37 | 37.01 | 93.53 |
| | GradNorm[12] | 82.45 | 79.85 | 19.23 | 96.77 | 48.99 | 90.67 | 69.40 | 81.72 | 55.02 | 87.25 |
| | ReAct[8] | 46.87 | 92.54 | 22.80 | 96.10 | 18.31 | 96.92 | 47.39 | 91.58 | 33.84 | 94.29 |
| | BATS(Ours) | **38.42** | **93.53** | 17.75 | 96.91 | 19.85 | 96.59 | **43.81** | **92.32** | **29.96** | **94.84** |
| CIFAR10 WRN | MSP[6] | 63.24 | 86.66 | 39.57 | 94.60 | 44.31 | 93.82 | 60.71 | 88.90 | 51.96 | 91.00 |
| | ODIN[7] | 61.13 | 82.49 | **12.79** | **97.61** | 12.49 | 97.50 | 61.13 | 80.18 | 36.89 | 89.45 |
| | Energy[5] | 56.05 | 86.63 | 17.58 | 96.99 | 28.44 | 95.29 | 61.74 | 85.68 | 40.95 | 91.15 |
| | GradNorm[12] | 88.55 | 49.14 | 41.25 | 90.68 | 91.02 | 48.94 | 90.83 | 46.28 | 77.91 | 58.76 |
| | ReAct[8] | 58.35 | 86.67 | 18.85 | 96.62 | 16.52 | 97.04 | 50.89 | 89.27 | 36.15 | 92.40 |
| | BATS(Ours) | **50.60** | **89.50** | 25.17 | 95.66 | **11.98** | **97.70** | **45.30** | **91.18** | **33.26** | **93.51** |
| CIFAR100 RN18 | MSP[6] | 81.79 | 77.80 | 68.32 | 83.92 | 82.51 | 75.73 | 85.12 | 73.36 | 79.44 | 77.70 |
| | ODIN[7] | **40.82** | **93.32** | 69.34 | 86.28 | 79.62 | 82.12 | 83.61 | 72.36 | 68.35 | 83.52 |
| | Energy[5] | 81.24 | 84.59 | 40.12 | 93.16 | 73.56 | 82.98 | 85.87 | 74.94 | 70.20 | 83.92 |
| | GradNorm[12] | 57.65 | 87.77 | **25.77** | **95.12** | 89.60 | 63.25 | 79.08 | 68.89 | 63.03 | 78.76 |
| | ReAct[8] | 70.28 | 88.25 | 45.62 | 91.02 | 55.57 | 89.32 | 61.01 | 87.57 | 58.12 | 89.04 |
| | BATS(Ours) | 61.48 | 90.63 | 44.41 | 91.27 | **52.68** | **90.04** | **52.36** | **89.72** | **52.73** | **90.42** |
| CIFAR100 WRN | MSP[6] | 78.43 | 77.74 | 61.33 | 87.46 | 81.69 | 72.69 | 85.07 | 75.46 | 76.63 | 78.34 |
| | ODIN[7] | **35.69** | **94.84** | 82.68 | 79.17 | 87.48 | 74.53 | 86.97 | 65.40 | 73.21 | 78.49 |
| | Energy[5] | 75.57 | 83.05 | **40.87** | **92.99** | 65.90 | 82.78 | 87.98 | 71.21 | 67.58 | 82.51 |
| | GradNorm[12] | 83.24 | 72.55 | 45.20 | 90.43 | 78.62 | 68.80 | 92.59 | 46.99 | 74.91 | 69.69 |
| | ReAct[8] | 72.94 | 86.89 | 42.07 | 91.97 | 60.87 | 85.90 | 84.18 | 76.22 | 65.02 | 85.25 |
| | BATS(Ours) | 71.01 | 87.50 | 41.93 | 91.97 | **57.01** | **88.04** | 80.46 | 78.42 | **62.60** | **86.48** |

## 5.3 Evaluation on CIFAR benchmarks

We further evaluate our method on CIFAR benchmarks and use CIFAR-10 and CIFAR-100 [38] as the in-distribution datasets respectively. Tab. 2 compares our method with the baseline methods and shows the OOD detection performance for each OOD test dataset and the average over the four datasets. We evaluate our method on the ResNet-18 (RN18) [42] and WideResNet-28-10 (WRN) [47]. The models are trained for 200 epochs with a batch size of 128. The starting learning rate is 0.1 and decays by a factor of 10 at epochs 100 and 150.

ODIN [7] performs the best in the baselines methods on CIFAR-10 with an FPR95 of 32.99% on ResNet-18. Our method outperforms ODIN by 3.03% and outperforms the simple baseline method MSP [6] by 27.14% in FPR95. As for using CIFAR-100 as the in-distribution dataset, ReAct [8] is the best baseline method. Our approach surpasses the ReAct by 5.39% in FPR95 on ResNet-18. Our method achieves the best performance on both CIFAR-10 and CIFAR-100. Our approach is also effective when using the WideResNet model, outperforming the existing methods.

## 5.4 Ablation studies

### 5.4.1 Rectifying the features of the early layers

In our experiments, we rectify the features of the penultimate layer (the layer before the fully connected layer), which is convenient and efficient. However, **what will happen if we rectify the features of the early layers with BATS?** The early layers refer to the layers close to the input [48]. In particular, the original ResNet-50 [42] consists of four residual blocks. Block1 is close to the input and Block4 is close to the output. In Tab. 3, we show the influence of applying feature rectification on the output of different blocks. Applying feature rectification to the early blocks (from Block1 to Block3) has little effect on the performance of OOD detection, while the last block plays a vital role. Applying feature rectification on all the blocks performs the best in our experiments, which is 0.99% higher than the "Block4" in FPR95 and 32.04% higher than "Without" in FPR95. Considering that the latest block has a more significant impact on the OOD detection performance than the other blocks, we just rectify the features of the penultimate layer with **BATS** for the simplicity of the method.

To find out why the last block has a significant influence on the OOD detection while the other blocks contribute little, we visualize the feature embeddings extracted by different blocks in ResNet-50 using t-SNE [49] in Fig. 4. We choose the iNaturalist as the OOD dataset and the ImageNet as the ID dataset. The features extracted by the early blocks of the ID and OOD examples are similar, which has little benefit in distinguishing the ID and OOD examples. In contrast, the last block can extract

Table 3: Ablation study of the influence of feature rectification on different blocks. "Without" means applying no rectification on any blocks. "Block1-4" means applying rectification on all blocks.

| Blocks | iNaturalist | | SUN | | Places | | Textures | | Average | |
|---|---|---|---|---|---|---|---|---|---|---|
| | FPR95 | AUROC | FPR95 | AUROC | FPR95 | AUROC | FPR95 | AUROC | FPR95 | AUROC |
| Without | 46.65 | 91.32 | 61.96 | 84.88 | 67.97 | 82.21 | 56.06 | 84.88 | 58.16 | 85.82 |
| Block1 | 49.72 | 90.73 | 62.67 | 84.62 | 68.30 | 82.03 | 55.62 | 85.04 | 59.08 | 85.61 |
| Block2 | 41.78 | 92.36 | 63.73 | 84.67 | 69.45 | 81.95 | 55.53 | 85.45 | 57.62 | 86.11 |
| Block3 | 40.76 | 92.55 | 58.37 | 86.56 | 64.78 | 83.82 | 51.45 | 86.77 | 53.84 | 87.43 |
| Block4 | **12.57** | **97.67** | 22.62 | 95.33 | 34.34 | 91.83 | 38.90 | 92.27 | 27.11 | 94.28 |
| Block1-2 | 43.63 | 91.92 | 63.22 | 84.61 | 69.28 | 81.84 | 53.67 | 85.72 | 57.45 | 86.02 |
| Block1-3 | 38.05 | 93.04 | 59.47 | 86.47 | 66.30 | 83.49 | 49.72 | 87.50 | 53.39 | 87.63 |
| Block1-4 | 12.76 | 97.54 | **21.15** | **95.51** | **33.01** | **91.91** | **37.55** | **92.54** | **26.12** | **94.38** |

perfectly separable features for the ID and OOD examples. This may be due to the fact that deep neural networks focus on similar general features (edges, lines, and colors) in the early layers and pay more attention to specific features related to classification in the late layers [48, 50]. The late layer can contribute more to the OOD detection than the early layer. See more in Appendix B.

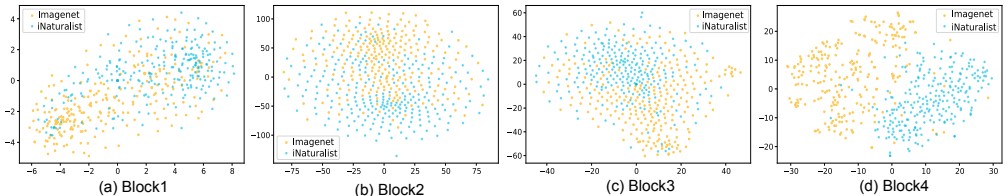

Figure 4: t-SNE visualizations. We illustrate the t-SNE plots for the features of in-distribution examples (ImageNet) and out-of-distribution examples (iNaturalist) from different blocks.

### 5.4.2 The influence of the hyperparameter

In Sec. 4.3, we theoretically analyze the bias-variance trade-off in our method. Our proposed **BATS** can reduce variance, which benefits OOD detection, but can also introduce a bias. Here, we empirically show the influence of the hyperparameter $\lambda$ in Fig. 5. As $\lambda$ tends to infinity, BATS approaches to the Energy Score (the horizontal lines). Very small $\lambda$ will damage the performance.

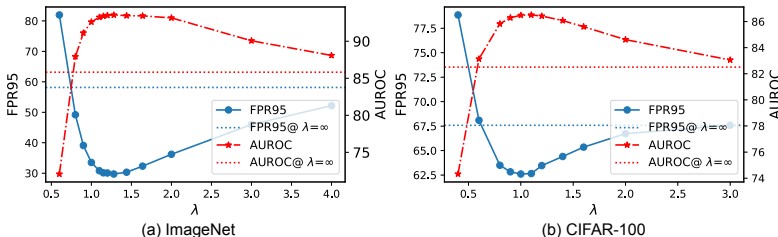

Figure 5: (a) The influence of the hyper-parameter $\lambda$ on the OOD detection on ImageNet. The model is ResNet-50. We illustrate the average performance on four OOD datasets. (b) The influence of the hyper-parameter $\lambda$ on the OOD detection on CIFAR-100. The model is WideResNet. The horizontal line indicates the OOD detection performance without feature rectification.

## 6 Conclusion

In this paper, we provide novel insight into the obstacle factor in OOD detection from the perspective of typicality and hypothesize that extreme features can be the culprit. We propose to rectify the features into the typical set and provide a concise and effective post-hoc approach **BATS** to estimating the feature's typical set. **BATS** can be applied to various OOD scores to boost the OOD detection performance. Theoretical analysis and ablations provide a further understanding of our approach. Experimental results show that our **BATS** can establish state-of-the-art OOD detection performance on the ImageNet benchmark, surpassing the previous best method by 5.11% in FPR and 1.43% in AUROC. We hope that our findings can motivate new research into the internal mechanisms of deep models and OOD detection and uncertainty estimation from the perspective of feature typicality.

**Limitations and societal impact.** This paper proposes to rectify the feature into its typical set to improve the detection performance against OOD data and provides a plug-and-play method with the assistance of BN. The limitation of our method can be that the BN layers are required in the model architecture in our approach. BN layers are widely used in convolutional neural networks to alleviate covariate shifts, but there are also architectures without BN. A set of training images can contribute to selecting the feature's typical set and alleviate this limitation. We also anticipate some other information in the model is conducive to selecting the feature's typical set and improving the post-hoc OOD detection performance. We leave this as future work. Although truncating features into a typical set can improve OOD detection, a potential negative impact of the proposed process is that it inherently introduces a bias and causes some information loss which may be important to the model in real-world scenarios.

## Acknowledgments

This work was supported in part by the Fundamental Research Funds for the Central Universities, by Alibaba Group through Alibaba Research Intern Program, and by the National Nature Science Foundation of China 62001146. Dr. Xie's research work is partially supported by the Interdisciplinary Intelligence SuperComputer Center of Beijing Normal University at Zhuhai.

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
