# Appendix: Boosting Out-of-distribution Detection with Typical Features

## A  Type I error and type II error in OOD detection

In the preliminary section, we provide a summary for the out-of-distribution detection from the perspective of hypothesis testing. As for the error of an OOD detection method, it can be evaluated from two dimensions. The mistaken rejection of an actually true null hypothesis $\mathcal{H}_0$ is the type I error. The significance level $\alpha$ is a predetermined scalar that bounds the type I error above:

$$\alpha >= P(\mathbf{x} \in \mathcal{R}|\mathcal{H}_0) = P(T(\mathbf{x}; f) \geq \gamma|\mathcal{H}_0) = P_0(T(\mathbf{x}; f) \geq \gamma).$$

By the Neyman–Pearson lemma [1], the threshold $\gamma$ is determined by solving the equation $\alpha = P_0(T(\mathbf{x}; f) \geq \gamma)$. For a given significance level, the goal is to minimize the type II error: the failure to reject a null hypothesis that is actually false. The probability of the type II error is denoted by

$$\beta = P(x \notin \mathcal{R}|\mathcal{H}_1) = P(T(\mathbf{x}; f) < \gamma|\mathcal{H}_1).$$

In the literature on OOD detection, the type II error is also denoted by "FPR$(1 - \alpha)$", which is short for "the false positive rate of OOD examples when the true positive rate for ID examples is $(1 - \alpha)\%$." In the experiments, we follow the notation FPR$(1 - \alpha)$.

In our paper, we mainly show the superiority of our method on different datasets in the metrics of FPR95 and AUROC. Here we illustrate the OOD detection performance at different significance levels (FPR(1-$\alpha$)) in Fig. 1. The horizontal axis represents the significance level for FPR ("0.95" means FPR95). Our method surpasses the existing methods at different significance levels on both the large scale dataset (ImageNet) and the small scale dataset (CIFAR-10).

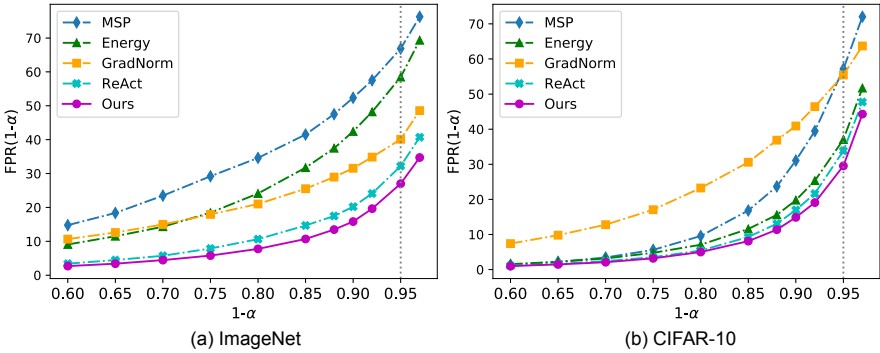

Figure 1: (a) The FPR(1-$\alpha$) for different methods on ImageNet (lower is better). The model is ResNet-50. We illustrate the average performance on four OOD datasets. The grey vertical line indicates the performance in FPR95. (b) The FPR(1-$\alpha$) for different methods on CIFAR-10. The model is ResNet-18.

36th Conference on Neural Information Processing Systems (NeurIPS 2022).

## B  The influence of the early layers

In the paper, we show that the early layers of the model can hardly distinguish the feature embeddings of the in-distribution examples (ImageNet-1k) and out-of-distribution examples (iNaturalist) for that the features of these examples extracted by the early layers are mixed up. Rectifying the features of the early layers contributes little to OOD detection. In this section, we illustrate the t-SNE visualization for the feature embeddings of in-distribution examples and other out-of-distribution examples (Places [2], SUN [3], and Textures [4]) from different blocks. Their t-SNE visualization results are similar. To be specific, the feature embeddings of the early blocks of the different datasets are similar, while the last block shows differences. In Tab. 3 in the main paper, we set the $\lambda$ for Block1 and Block2 as 5, the $\lambda$ for Block3 as 2, and the $\lambda$ for Block4 as 1, for that restricting the features of the early layers may have a negative impact on the late layers.

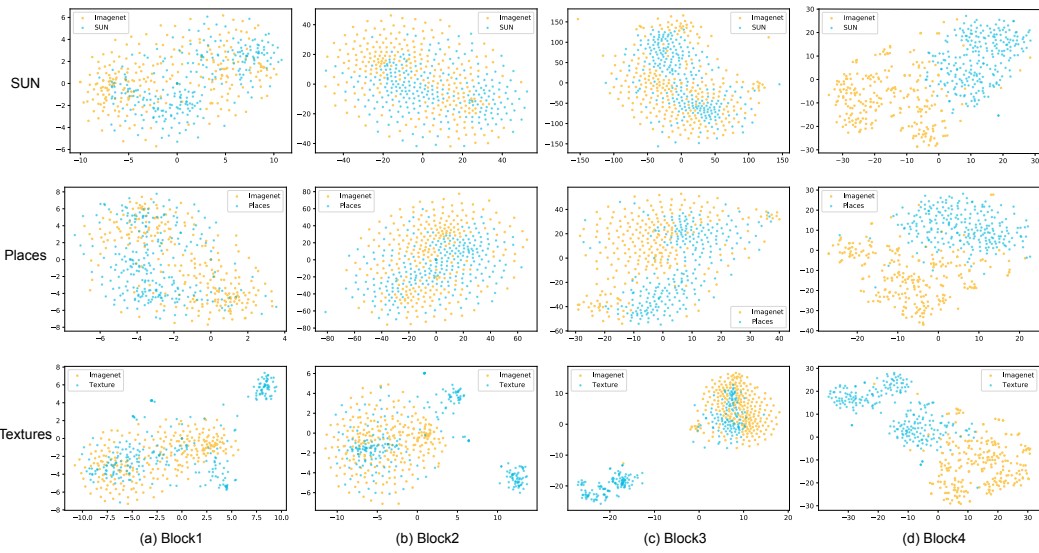

Figure 2: t-SNE visualization for the feature embeddings of in-distribution examples and out-of-distribution examples from different blocks. The model we used is ResNet-50.

## C  Energy and density in the classifier

To make our paper self-contained, we provide some details for the energy and density in the classifier with reference to the previous works [5–7].

LeCun et al. [8] show that any probability density $p(x)$ for $x$ can be expressed as

$$p(x) = \frac{\exp(-E(x))}{Z},$$ (1)

where $E(x)$ represents the energy of $x$ and is modeled by neural network, $Z = \int \exp(-E(x))dx$ is the normalizing factor which is also known as the partition function.

Similarly, $p(x, y)$ can be defined as follows:

$$p(x, y) = \frac{\exp(-E(x, y))}{\tilde{Z}},$$ (2)

where $\tilde{Z} = \int \sum_y \exp(-E(x, y))dx$.

Thus we also get $p(y|x)$ expressed by $E(x)$ and $E(x, y)$:

$$p(y|x) = \frac{p(x, y)}{p(x)} = \frac{\exp(-E(x, y)) \cdot Z}{\exp(-E(x)) \cdot \tilde{Z}}.$$ (3)

We denote $f$ as a classification neural network. Let $x$ be a sample. Then $f(x)[k]$ represents the $k^{th}$ output of the last layer and $p(y|x)$ can be defined as:

$$p(y|x) = \frac{\exp(f(x)[y])}{\sum_{k=1}^{n} \exp(f(x)[k])}, \tag{4}$$

where $n$ represents total possible classes. From Eq. (3) and (4), we define two energy functions as follows:

$$\begin{cases} E(x, y) = -\log(\exp(f(x)[y])), \\ E(x) = -\log(\sum_{k=1}^{n} \exp(f(x)[k])). \end{cases} \tag{5}$$

And thus $Z$ can be expressed as:

$$Z = \int_x \exp(-E(x))dx = \int_x \exp(\log(\sum_y \exp(f(x)[y])))dx = \int_x (\sum_y \exp(f(x)[y]))dx = \tilde{Z}. \tag{6}$$

From Eq. (5) and Eq. (1), the marginal density $p(x)$ for $x$ can be expressed by the output of the classifier as:

$$p(x) = \frac{\sum_{k=1}^{n} \exp(f(x)[k])}{Z}, \tag{7}$$

where $Z$ is independent to $\mathbf{x}$.

## D    Proofs of section 4.3

In this section, we prove the main results in Section 4.3. Recall the layer structure in Eq. (3) in the main paper:

$$\mathbf{z}' \to \text{BN}(\mathbf{z}'; \mu, \sigma) \text{ or } \text{TrBN}(\mathbf{z}'; \mu, \sigma, \lambda) \to \text{ReLU} \to \mathbf{z}, \tag{8}$$

where $\mathbf{z}'$ is the feature vector extracted from the penultimate layer of $g$. We denote

$$\mathbf{z}_1 = \text{BN}(\mathbf{z}'; \mu, \sigma) \quad \text{and} \quad \bar{\mathbf{z}}_1 = \text{TrBN}(\mathbf{z}'; \mu, \sigma). \tag{9}$$

Suppose $\mathbf{z}'$ is a Gaussian variable. Then $\mathbf{z}_1 \sim N(\mu, \sigma^2)$ and $\bar{\mathbf{z}}_1$ follows a Rectified Gaussian distribution with lower bound $\mu - \lambda\sigma$ and upper bound $\mu + \lambda\sigma$. The cdf of $\bar{\mathbf{z}}_1$ is

$$F^R(\mathbf{z}'|\mu, \sigma^2) = \begin{cases} 0, & \text{if } \mathbf{z}' < a; \\ \Phi(\mathbf{z}'; \mu, \sigma^2), & \text{if } a \leq \mathbf{z}' < b; \\ 1, & \text{if } b \leq \mathbf{z}', \end{cases} \tag{10}$$

where $\Phi(\mathbf{z}'; \mu, \sigma^2)$ the cdf of a normal distribution with mean $\mu$ and variance $\sigma^2$. According to [9],

$$\mathbb{E}(\bar{\mathbf{z}}_1) = \mu \quad \text{and} \quad \text{Var}(\bar{\mathbf{z}}_1) = \sigma^2 C(\lambda), \tag{11}$$

where

$$C(\lambda) = \text{erf}(\frac{\lambda}{\sqrt{2}}) - \frac{\sqrt{2}}{\sqrt{\pi}} \lambda \exp(-\frac{\lambda^2}{2}) + \lambda^2(1 - \text{erf}(\frac{\lambda}{\sqrt{2}})), \tag{12}$$

and $\text{erf}(x) = (2/\sqrt{\pi}) \int_0^x \exp(-t^2)dt$ is the Gauss error function. It is easy to see

$$C(0) = 0 \quad \text{and} \quad C'(\lambda) = 2\lambda \cdot \text{erf}(\frac{\lambda}{\sqrt{2}}) > 0. \tag{13}$$

In addition,

$$\lambda^2(1 - \text{erf}(\frac{\lambda}{\sqrt{2}})) = \lambda^2 \frac{2}{\sqrt{\pi}} \int_{\lambda/\sqrt{2}}^{+\infty} \exp(-t^2)dt \tag{14}$$

$$\leq \frac{4}{\sqrt{\pi}} \int_{\lambda/\sqrt{2}}^{+\infty} t^2 \exp(-t^2)dt \to 0, \quad \lambda \to +\infty.$$

Therefore, as $\lambda$ tends to $+\infty$,

$$\text{erf}(\frac{\lambda}{\sqrt{2}}) \to 1, \quad \frac{\sqrt{2}}{\sqrt{\pi}} \lambda \exp(-\frac{\lambda^2}{2}) \to 0, \quad \lambda^2(1 - \text{erf}(\frac{\lambda}{\sqrt{2}})) \to 0. \tag{15}$$

We obtain that $C(\lambda) \to 1$ as $\lambda \to +\infty$.

Next we deal with the bias term. To proceed further, we need more notations as follow:

$$\mathbf{z} = \text{ReLU}(\text{BN}(\mathbf{z}'; \mu, \sigma)) \quad \text{and} \quad \bar{\mathbf{z}} = \text{ReLU}(\text{TrBN}(\mathbf{z}'; \mu, \sigma)). \tag{16}$$

Then we know that $\mathbf{z}$ follows a Rectified Gaussian distribution with lower bound 0 and upper bound $+\infty$ and $\bar{\mathbf{z}}$ is a Rectified Gaussian variable with lower bound 0 and upper bound $\mu + \lambda\sigma$. According to [9], their expectations are

$$\mathbb{E}(\mathbf{z}) = \mu + \sigma\Big(\frac{1}{\sqrt{2\pi}}\big(\exp(-\frac{\mu^2}{2\sigma^2})\big) - \frac{\mu}{2\sigma}(1 + \text{erf}(-\frac{\mu}{\sqrt{2}\sigma}))\Big), \tag{17}$$

and

$$\mathbb{E}(\bar{\mathbf{z}}) = \mu + \sigma\Big(\frac{1}{\sqrt{2\pi}}\big(\exp(-\frac{\mu^2}{2\sigma^2}) - \exp(-\frac{\lambda^2}{2})\big) \tag{18}$$
$$-\frac{\mu}{2\sigma}(1 + \text{erf}(-\frac{\mu}{\sqrt{2}\sigma})) + \frac{\lambda}{2}(1 - \text{erf}(\frac{\lambda}{\sqrt{2}}))\Big).$$

Therefore the bias term is

$$\mathbb{E}(\bar{\mathbf{z}}) - \mathbb{E}(\mathbf{z}) = \sigma\Big(-\exp(-\frac{\lambda^2}{2}) + \frac{\lambda}{2}(1 - \text{erf}(\frac{\lambda}{\sqrt{2}}))\Big). \tag{19}$$

In addition,

$$\frac{\lambda}{2}(1 - \text{erf}(\frac{\lambda}{\sqrt{2}})) = \frac{\lambda}{2}\frac{2}{\sqrt{\pi}}\int_{\lambda/\sqrt{2}}^{+\infty}\exp(-t^2)\mathrm{d}t \tag{20}$$
$$\leq \frac{\sqrt{2}}{\sqrt{\pi}}\int_{\lambda/\sqrt{2}}^{+\infty}t\exp(-t^2)\mathrm{d}t \to 0 \quad \text{as} \quad \lambda \to +\infty.$$

Then we obtain that

$$\text{Bias} = \mathbb{E}(\bar{\mathbf{z}}) - \mathbb{E}(\mathbf{z}) \to 0 \quad \text{as} \quad \lambda \to +\infty. \tag{21}$$

# E  Experiments details

## E.1  Details for metrics

**FPR95:** the false positive rate of OOD (negative) examples when the true positive rate of in-distribution (positive) examples is as high as 95%. The **t**rue **p**ositive **r**ate (TPR) can be computed as:

$$TPR = \frac{TP}{(TP + FN)}, \tag{22}$$

where TP denotes the true positive (correctly identify the in-distribution examples as in-distribution examples) and FN denotes the False Negative (incorrectly identity the in-distribution examples as out-of-distribution examples). The **f**alse **p**ositive **r**ate (FPR) can be computed as:

$$FPR = \frac{FP}{(FP + TN)}, \tag{23}$$

where FP denotes the false positive (incorrectly identify the out-of-distribution examples as in-distribution examples) and TN denotes the true negative (correctly identify the out-of-distribution examples as out-of-distribution examples).

**AUROC:** the area under the receiver operating characteristic curve (ROC) which is the plot of TPR vs FPR. If FPR = 0 and TPR = 1, it means that this is a perfect OOD detector, which identify all examples correctly. If FPR=1 and TPR=0, this is a terrible detector that can not make any correct prediction. The closer the area under the ROC curve is to 1, the better the performance of the detector.

### E.2 Details for datasets

#### E.2.1 CIFAR OOD detection

We use the CIFAR-10 and CIFAR-100 [10] as the in-distribution examples respectively. CIFAR-10 consists of 60,000 images in the shape of $3 \times 32 \times 32$, including 10 categories (aircraft, cars, birds, cats, deer, dogs, frogs, horses, boats, and trucks). CIFAR-100 contains 100 categories of images, and each category has 600 images in the shape of $3 \times 32 \times 32$. We evaluate our approach on four common OOD datasets. We set $\lambda = 3$ in our approach for CIFAR-10 and $\lambda = 1.5$ and for CIFAR-100 on ResNet-18. As for WideResNet-28-10, we set $\lambda = 0.7$ for CIFAR-10 and $\lambda = 1.0$ for CIFAR-100. During the evaluation, all images are resized to $3 \times 32 \times 32$.

**SVHN** [11]: Street View House Number (SVHN) consists of the house numbers extracted from Google Street View images. We use the entire of its test set as OOD examples (26032 images).

**Tiny ImageNet** [12]: Similar to ImageNet, Tiny ImageNet is an image classification dataset, which contains 200 categories, and each category contains 50 test images. We randomly crop the images to $3 \times 32 \times 32$.

**LSUN** [13]: LSUN is a scene understanding dataset, which mainly includes scene images of bedrooms, fixed houses, living rooms, classrooms, etc. We randomly sample 10000 images as out-of-distribution examples and resize the images to $3 \times 32 \times 32$.

**Textures** [4]: Describable Textures Dataset (DTD) is a texture dataset, including 5640 images, which can be divided into 47 categories according to human perception. We use the entire Textures dataset for evaluation.

#### E.2.2 Large-scale OOD detection

We use the subsets from the following datasets as OOD examples and follow the setting in [14] and [15]. The subsets are curated to be disjoint from the ImageNet-1k labels. We set $\lambda = 1.25$ in our approach for ResNet-50 and DenseNet-121, and $\lambda = 0.4$ for MobileNet-V2. During the evaluation, all images are resized to $3 \times 224 \times 224$. All models here use the softplus non-linearity ($\beta = 35$), which can be expressed as the expectation of ReLU in a neighborhood [16] and provide more robust features.

**iNaturalist** [17]: iNaturalist contains 675,170 training and validation images from 5089 natural fine-grained categories, including 13 major categories such as plants, insects, birds, and mammals. We randomly sample 10000 images that are disjoint from ImageNet-1k for evaluation.

**Places** [2]: Places is a scene image dataset, which contains 10 million pictures and more than 400 different types of scene environments. We randomly sample 10000 images that are disjoint from ImageNet-1k for evaluation.

**SUN** [3]: The Scene UNderstanding (SUN) contains 397 well-sampled categories to evaluate the performance of scene recognition algorithms. We randomly sample 10000 images that are disjoint from ImageNet-1k for evaluation.

**Textures** [4]: Describable Textures Dataset (DTD) is a texture dataset, including 5640 images, which can be divided into 47 categories according to human perception. We use the entire Textures dataset for evaluation.

### E.3 Hardware

Our experiments are implemented by PyTorch [18] and runs on RTX-2080TI.

## F Boosting the OOD detection on the robust classifiers

The models used in our experiments (Tab. 1 in the main paper) are standard pre-trained. Salman et al. [19] show that adversarially robust models with less accuracy often perform better than their standard-trained counterparts in transfer learning. In Fig. 3 we evaluate the OOD detection performance on eight adversarially pre-trained ResNet-50 trained with $\ell_2$ perturbation of different strength $\epsilon$ [19] on the ImageNet benchmark. The horizontal axis represents the perturbation strength in training the

model, e.g. "0.05" represents the robust model trained with $\ell_2$ perturbation $\epsilon = 0.05$. The strength of perturbation has a great influence on GradNorm [15], but little influence on other methods. Our BATS surpasses all the existing methods on different robust models.

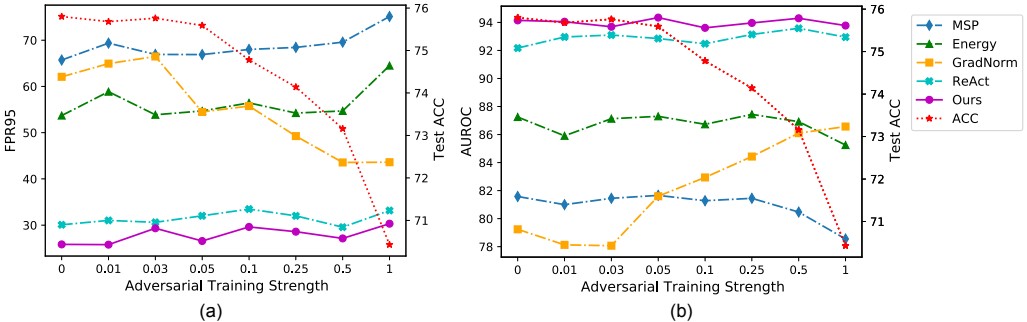

Figure 3: (a) The FPR95 for different methods on ImageNet benchmark (lower is better) on different robust models. We illustrate the average performance on four OOD datasets. The red dotted line indicates the test accuracy of the robust model. (b) The AUROC for different methods (higher is better).

# G    Related literature to OOD detection

Deep neural networks have been widely applied in various fields [20–23]. OOD detection has received wide attention because it is critical to ensuring the reliability and safety of deep neural networks. The literature related to OOD detection can be broadly grouped into the following themes. Our paper briefly reviews the literature related to post-hoc detection methods. Here, we provide a more comprehensive review.

**Post-hoc Detection Methods.** Post-hoc methods focus on improving the OOD uncertainty estimation by utilizing the pre-trained classifiers rather than retraining a model, which is beneficial for adopting OOD detection in real-world scenarios and large-scale settings. In this paper, we mainly focus on the post-hoc OOD detection methods. Hendrycks and Gimpel [24] observe that the maximum softmax probability of In-Distribution Examples (ID) can be higher than the Out-of-Distribution (OOD) samples and provide a simple baseline for OOD detection. ODIN [25] introduces a large sufficiently temperature factor and input perturbation to separate the ID and OOD examples. Liu et al. [7] analyze the limitations of softmax function in OOD detection and propose to use energy score as an indicator. The examples with high energy are considered as OOD examples, and vice versa. Wang et al. [26] propose to use joint energy score which take labels into consideration to enhance the OOD detection. ReAct [14] hypothesize that the OOD examples can trigger the abnormal activation of the model, and propose to clamp the activation value larger than the threshold value to improve the detection performance. Lee et al. [27] use the mixture of Gaussians to model the distribution of feature representations and propose using the feature-level Mahalanobis distance instead of the model output. GradNorm [15] shows that the gradients of the categorical cross-entropy loss contains useful information for OOD detection.

**Confidence Enhancement Methods.** To enhance the sensitivity to OOD examples, some methods propose to introduce the adversarial examples into the training process. Hein et al. [28] endow low confidence predictions to the examples far away from the training data through an adversarial training optimization technique. Moreover, Bitterwolf et al. [29] enforce low confidence in an l2 ball around the OOD examples. Proper data augmentation also contributes to OOD uncertainty estimation [30–32]. Some methods take advantage of a set of collected OOD examples to enhance the uncertainty estimation, which are named outlier exposure methods [33–35]. The correlations between the collected and real OOD examples can largely affect the performance of outliers exposure methods [36].

**Density-based Methods.** Directly estimating the density of the examples can be a natural approach, this kind of methods explicitly model the distribution of ID examples with probabilistic models and distinguish the OOD/ID examples through the likelihood [37–40]. However, some works show that

the probabilistic models may assign higher likelihood to OOD examples than ID examples and fail to distinguish OOD/ID examples [41, 42].

## H   Test accuracy

In this section, we show that with a proper hyper-parameter $\lambda$, our feature rectification method can slightly improve the test accuracy of the pre-trained models both on the clean images and the corrupted images. Here we set $\lambda = 3$. As shown in Tab. 1, we evaluate the test accuracy of the normal pre-trained models and the pre-trained models with our feature rectification method on the clean images and the corrupted images. We choose some image corruption methods used in [43], including salt-and-pepper noise (SP(0.2)), cropout (Crop(0.8)), JPEG compression (JPEG(50)), Gaussian blur (GB) and Gaussian noise (GN). Fig. 4 shows some examples that can be classified correctly by our feature rectified ResNet-50 but classified wrongly by the original ResNet-50.

Table 1: Test accuracy on ImageNet with the pre-trained ResNet-50 and DenseNet-121. Our method rectifies the feature vector of the model and performs well on both the clean images and the corrupted images. The best results are in bold.

| Model | Method | Vanilla | SP(0.2) | Crop(0.6) | JPEG(50) | GB(2) | GB(3) | GN(0.5) | GN(1) |
|-------|--------|---------|---------|-----------|----------|-------|-------|---------|-------|
| RN50  | Normal | 74.548 | 40.436 | 65.488 | 58.426 | 52.762 | 49.022 | 28.638 | 10.864 |
|       | Ours   | **74.610** | **40.506** | **65.770** | **58.454** | **52.920** | **49.312** | **28.852** | **11.014** |
| DN121 | Normal | 71.956 | 43.078 | 63.150 | 61.298 | 50.362 | 46.980 | 40.600 | 25.786 |
|       | Ours   | **72.050** | **43.146** | **63.498** | **61.332** | **50.520** | **47.014** | **40.618** | **25.852** |

## I   BATS on other detection methods

In our paper, we provide a concise and effective approach **BATS** to improve the performance of the existing OOD detection methods. We mainly show the effectiveness of applying our feature rectification (BATS) on Energy Score [7]. In Tab. 2 we show that out method is compatible with various OOD detection test statistics (including output-based methods [24, 7, 25] and gradient-based method [15]) and can bring improvements to different methods. Applying our **BATS** on GradNorm can even achieve better performance than "Energy+**BATS**" but this method needs to derive the gradients of the model which cost more than "Energy+**BATS**".

OOD detection methods hope to assign higher scores to the in-distribution (ID) examples and lower scores to the out-of-distribution (OOD) examples. The advanced detection method (GradNorm [15]) can assign better scores to ID and OOD examples than the simple baseline method (MSP [24]). However, there still exists an overlap in the distribution of the scores. As shown in Fig. 5, our BATS reduces the variance of the scores and makes the scores of the ID and OOD examples more separable, which can improve the performance of the OOD detection methods. We think combining our method with a better OOD score can achieve better performance. "BATS+Energy" has already achieved state-of-the-art performance on large-scale and small-scale benchmarks.

## J   The feature distribution on different channels

Fig. 6 illustrates the feature distribution on different channels of in-distribution examples (ImageNet) and the out-of-distribution examples on ResNet-50. These features are extracted by the last convolution block. We name the region where features are concentrated as the typical set of features. These regions receive more attention during training, and the model is more familiar with the features in these regions than those in extreme regions. For better visual presentation, we illustrate features before ReLU.

## K   Detection on natural adversarial examples

In our paper, we follow the settings of the existing research, choosing iNaturalist, Places, SUN and Textures as out-of-distribution datasets and choosing ImageNet-1k as the in-distribution dataset. Here

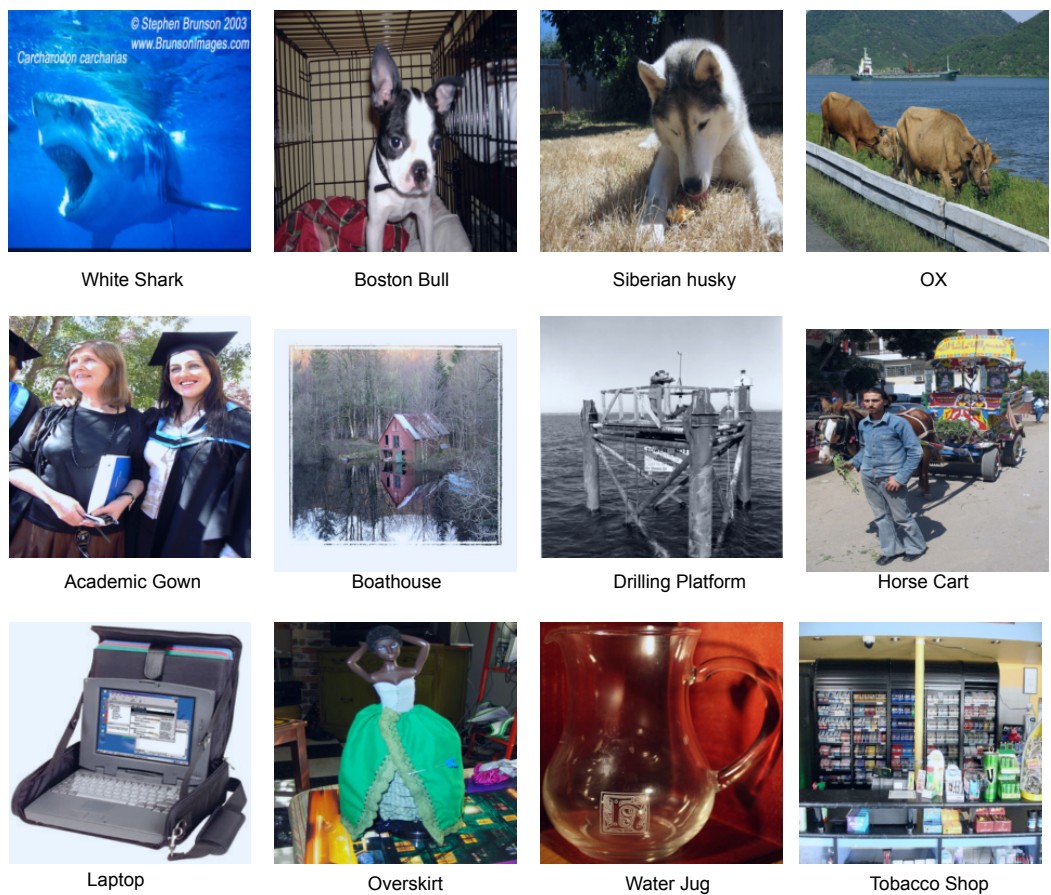

| White Shark | Boston Bull | Siberian husky | OX |
| Academic Gown | Boathouse | Drilling Platform | Horse Cart |
| Laptop | Overskirt | Water Jug | Tobacco Shop |

Figure 4: We illustrate some examples in ImageNet that can be classified correctly by the feature rectified ResNet-50 but classified wrongly by the original ResNet-50.

Table 2: Applying feature rectification (**BATS**) and ReAct [14] on different OOD detection methods. The best results are in bold.

| Model | Method | iNaturalist | | SUN | | Places | | Textures | | Average | |
|---|---|---|---|---|---|---|---|---|---|---|---|
| | | FPR95 | AUROC | FPR95 | AUROC | FPR95 | AUROC | FPR95 | AUROC | FPR95 | AUROC |
| RN50 | MSP | 51.44 | 88.17 | 72.04 | 79.95 | 74.34 | 78.84 | 54.90 | 78.69 | 63.18 | 81.41 |
| | MSP+ReAct | 44.90 | 91.68 | 60.86 | 86.22 | 64.95 | 84.48 | 62.06 | 85.46 | 58.19 | 86.96 |
| | MSP+**BATS** | 35.79 | 93.56 | 56.97 | 88.08 | 63.24 | 85.35 | 55.14 | 87.93 | 52.79 | 88.73 |
| | ODIN | 41.07 | 91.32 | 64.63 | 84.71 | 68.36 | 81.95 | 50.55 | 85.77 | 56.15 | 85.94 |
| | ODIN+ReAct | 32.10 | 93.84 | 45.14 | 90.34 | 52.48 | 87.92 | 45.07 | 87.95 | 43.70 | 90.01 |
| | ODIN+**BATS** | 25.43 | 95.44 | 40.12 | 92.28 | 50.57 | 88.87 | 36.67 | 92.42 | 38.20 | 92.25 |
| | Energy | 46.65 | 91.32 | 61.96 | 84.88 | 67.97 | 82.21 | 56.06 | 84.88 | 58.16 | 85.82 |
| | Energy+ReAct | 17.77 | 96.70 | 25.15 | 94.34 | 34.64 | 91.92 | 51.31 | 88.83 | 32.22 | 92.95 |
| | Energy+**BATS** | 12.57 | 97.67 | 22.62 | 95.33 | 34.34 | 91.83 | 38.90 | 92.27 | 27.11 | 94.28 |
| | GradNorm | 23.73 | 93.97 | 42.81 | 87.26 | 55.62 | 81.85 | 38.15 | 87.73 | 40.08 | 87.70 |
| | GradNorm+ReAct | 12.95 | 97.74 | 26.41 | 94.85 | 38.44 | 91.70 | 29.55 | 93.78 | 26.84 | 94.52 |
| | GradNorm+**BATS** | 10.01 | 98.23 | 18.87 | 96.42 | 32.45 | 92.78 | 24.79 | 95.28 | 21.53 | 95.68 |
| MNet | MSP | 63.09 | 85.71 | 79.67 | 76.01 | 81.47 | 75.51 | 75.12 | 76.49 | 74.84 | 78.43 |
| | MSP+ReAct | 65.42 | 86.90 | 81.09 | 76.09 | 81.68 | 75.68 | 69.93 | 81.34 | 74.53 | 80.00 |
| | MSP+**BATS** | 49.77 | 90.60 | 70.75 | 80.66 | 74.66 | 78.45 | 57.61 | 85.61 | 63.20 | 83.83 |
| | ODIN | 45.61 | 91.33 | 63.03 | 83.44 | 70.01 | 80.85 | 52.45 | 85.61 | 57.78 | 85.31 |
| | ODIN+ReAct | 41.90 | 92.36 | 68.29 | 82.82 | 71.96 | 81.00 | 43.37 | 89.76 | 56.38 | 86.49 |
| | ODIN+**BATS** | 29.15 | 94.66 | 58.54 | 85.38 | 65.60 | 82.24 | 35.96 | 91.42 | 47.31 | 88.43 |
| | Energy | 49.52 | 91.10 | 63.06 | 84.42 | 69.24 | 81.42 | 58.16 | 84.88 | 60.00 | 85.46 |
| | Energy+ReAct | 37.08 | 93.41 | 53.13 | 86.04 | 54.15 | 83.31 | 42.45 | 89.42 | 46.70 | 88.05 |
| | Energy+**BATS** | 31.56 | 94.33 | 41.68 | 90.21 | 52.43 | 86.26 | 38.69 | 90.76 | 41.09 | 90.39 |
| | GradNorm | 33.70 | 92.46 | 42.15 | 89.65 | 56.56 | 83.93 | 34.95 | 90.99 | 41.84 | 89.26 |
| | GradNorm+ReAct | 25.85 | 95.03 | 38.94 | 91.42 | 52.94 | 86.74 | 18.85 | 95.75 | 34.15 | 92.24 |
| | GradNorm+**BATS** | 21.51 | 95.89 | 30.97 | 93.19 | 46.94 | 88.08 | 17.71 | 95.97 | 29.28 | 93.28 |

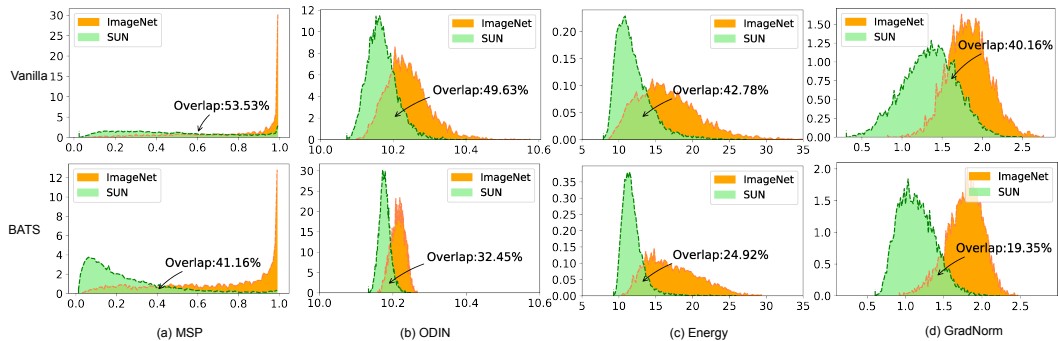

Figure 5: We illustrate different OOD scores for ID (ImageNet) and OOD (SUN) examples. "Vanilla" means the original OOD detection method, and "BATS" means applying our BATS to the detection method. BATS reduces the variance of the scores and reduces the overlap between the distribution of ID and OOD examples.

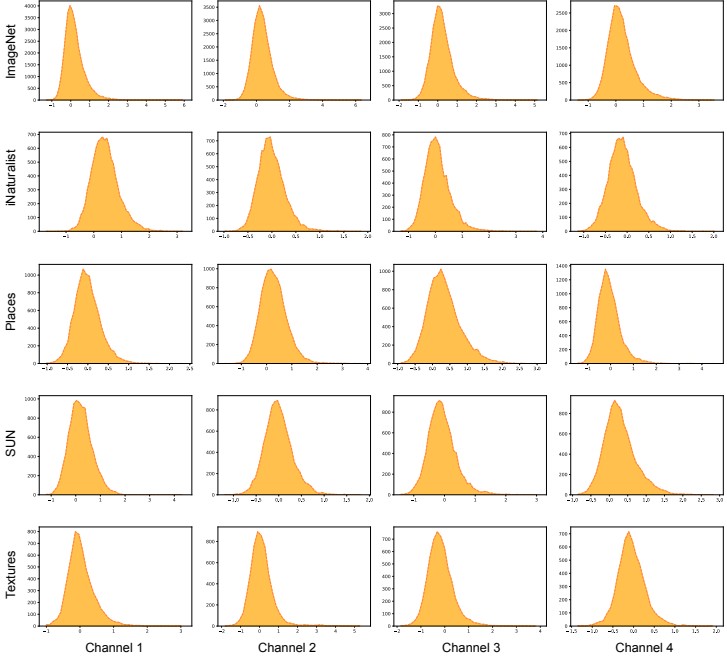

Figure 6: The feature distribution on different channels of in-distribution examples (ImageNet) and the out-of-distribution examples on ResNet-50. We randomly choose four channels of the features extracted by the penultimate layer.

we consider a much more challenging task: detecting natural adversarial examples [44]. Hendrycks et al. [44] introduce natural adversarial examples ImageNet-O, which are naturally occurring examples in the real world but significantly degrade the deep model performance. We use the ImageNet-O [44] which contains anomalies of unforeseen classes as the out-of-distribution examples. Fig. 7 shows that our method surpasses the existing methods by a large margin in FPR(1-$\alpha$) with different significance levels. Our method significantly improve the AUROC from 56.68% to 64.48%.

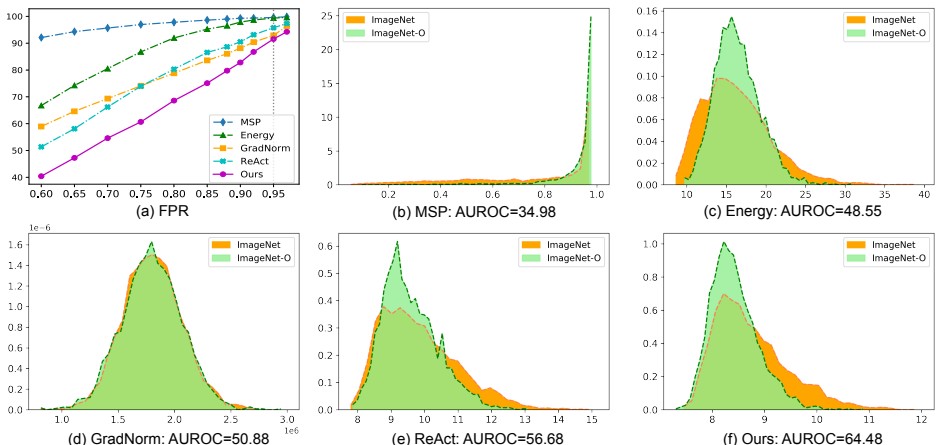

Figure 7: (a) The FPR(1-$\alpha$) for different method on Imagenet (lower is better). The model is ResNet-50 and the OOD dataset is ImageNet-O. (b)-(f) The frequency histogram of the scores for ImageNet and ImageNet-O.

## L   Class activation mapping

In this section, we use the Smooth Grad-CAM++[45] to generate the heat map for different images. As shown in Fig. 8, the heat map of our rectified model aligns better with the objects in the image than that of the original model. We use the pre-trained ResNet-50 in PyTorch [18]. The heat map of our rectified model for Fig. 8(b) (the mud turtle) shows that the head of the turtle dominates the decision while the original model pays more attention to the neck of the turtle. The rectified model takes more object-relevant parts into consideration, which may contribute to its slightly better test accuracy (in Appendix H).

## M   Additional Analysis for Performance Degradation Case in Tab.2

In Tab. 2 in the main paper, we show that the average performance of our **BATS** surpasses the baseline method Energy, but the performance degrades in the case using Tiny-Imagenet as the OOD dataset. We hypothesize that this performance degradation is due to the bias introduced by BATS. By truncating the features, BATS can reduce the variance of the in-distribution examples which benefits the estimation of the reject region but inherently cause some information loss which may reduce the performance of the pre-trained models.

To validate our hypothesis, we tune the bias-variance trade-off by the hyperparameter $\lambda$. As shown in Fig. 9, BATS can indeed reduce the variance of the OOD scores. With a proper $\lambda$, BATS can reduce the overlap between the ID and OOD examples and reduce the FPR95, while a small $\lambda$ hinders the performance of OOD detection. For example, using larger $\lambda = 8$, BATS can achieve better FPR95 performance 15.10% on detecting Tiny-Imagenet using ResNet-18, which is 2.65% better than $\lambda = 3$ in our Tab. 2 in the main paper. For the practicability of our method, we set the same hyperparameter to test different OOD datasets, without adjusting for specific OOD datasets.

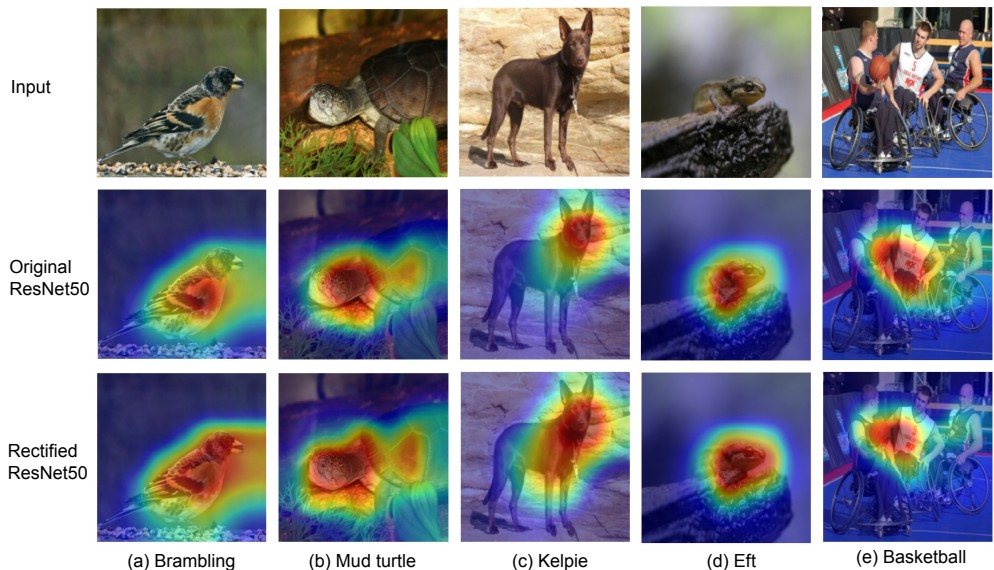

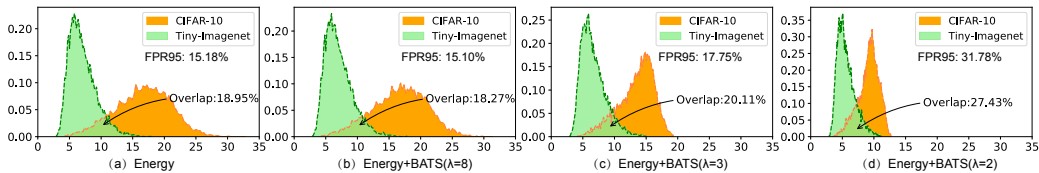

Figure 8: We draw the heat maps to explain which parts of the image dominate the model decision through Smooth Grad-CAM++[45]. The heat map of our rectified model for each image aligns better with the objects in the image than that of the original model.

Figure 9: The distribution of the scores for ID (CIFAR-10) and OOD examples (Tiny-ImageNet) on ResNet-18. Choosing a proper $\lambda$, BATS can reduce the overlap between the ID and OOD examples and reduce the FPR95, while a small $\lambda$ hinders the performance of OOD detection.

## N  The difference between BATS and ReAct

The similarity between our **BATS** and ReAct is that these methods are used to improve the performance of the existing OOD scores. As follows, we discuss the difference between **BATS** and ReAct from three aspects.

First, the motivation between our **BATS** and ReAct is different. ReAct hypothesizes that the mean activation of OOD data has significantly larger variations across units and is biased towards having sharp positive values, while the activation of the ID data is well-behaved with a near-constant mean and standard deviation. Thus, ReAct thinks that the truncation can rectify the activation of the OOD examples and preserve the activation of in-distribution data. However, as shown in Fig. 10, the mean activation of OOD data does not always have significantly larger variations than the ID data, which means this hypothesis does not always hold. The distribution of the deep features after batch normalization is consistent with the Gaussian distribution. Our **BATS** hypothesizes that deep models may be hard to model the extreme features but can provide reliable estimations on the typical features. This is because extreme features are exposed to the training process with a low probability. We propose to rectify the features into the typical set and calculate the OOD scores with the typical features.

Second, the mathematical analysis between our BATS and ReAct is different. ReAct theoretically analyze that if the OOD activations are more positively skewed, their operation reduces mean OOD activations more than ID activations. We analyze the benefit of **BATS** from the perspective of the bias-variance trade-off. BATS can reduce the variance of the deep features, which contributes to

constraining the uncertainty of the test static $T(x; f)$ and improving the estimation accuracy of the reject region. Our method hopes to estimate the reject region better, and we do not assume whether OOD data is positively skewed.

Third, our method surpasses the ReAct in both the large-scale benchmark (ImageNet) and the small-scale benchmark (CIFAR).

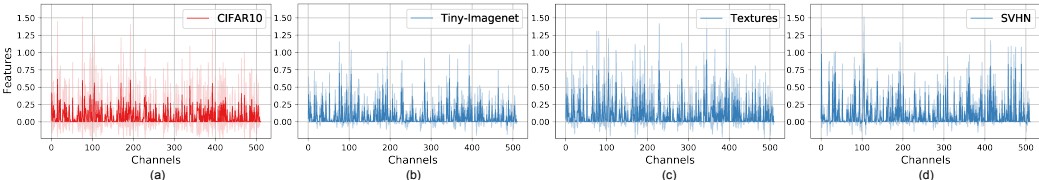

Figure 10: The distribution of the features of the in-distribution dataset (a) and out-of-distribution datasets (b-d) on different channels. We use the WideResNet-28-10 to extract the features. The mean and standard deviation are shown by the solid line and shaded area, respectively. Compared to other datasets, the mean value of the features in different channels of the Tiny-Imagenet is smaller and has a smaller standard deviation.

## O   Selecting features' typical set without assistance of BN

In our paper, we mainly analyze that rectifying the features in the typical set can improve the performance of the existing OOD scores. We provide a concise and effective method to select the typical set with the assistance of the BN layers and achieve a state-of-the-art performance among post-hoc methods on a suite of OOD detection benchmarks.

Here, we provide another method to select the features' typical set, which directly uses a set of training images to estimate the mean $\mu$ and the standard deviation $\sigma$ of the features (extracted by the penultimate layer of the model) at each dimension. Then we rectify the features into the interval $[\mu - \lambda * \sigma, \mu + \lambda * \sigma]$ and use these typical features to calculate the OOD scores. We named this method as **T**ypical **F**eature **E**stimated **M**ethod (TFEM). This method does not require the BN layers in the model but needs to use a set of training images.

In Tab. 3, we compare the OOD detection performance of the OOD detection methods with and without our TFEM. In this experiment, we randomly choose 1500 images from the training dataset of the ImageNet. The $\lambda$ is set to 1. The experiment is performed on the ImageNet benchmark. The models are pre-trained ResNet-50 and ViT. ViT (Vision Transformer) [46] is a transformer-based image classification model which treats images as sequences of patches and does not have BN layers. We use the officially released ViT-B/16 model, which is pre-trained on ImageNet-21K and fine-tuned on ImageNet-1K. Rectifying the features into the typical set with TFEM can greatly improve the performance of the existing OOD detection methods both on the model with BN layers (ResNet-50) and the model without BN layers (ViT).

This experiment demonstrates the effectiveness of the typical features in OOD detection, which is consistent with the analysis in our paper. We believe there exists a method that can estimate the features' typical set better. In this paper, BATS has already established state-of-the-art performance on both the large-scale and small-scale OOD detection benchmarks.

## P   Comparison between BATS and two latest detection methods

In this section, we compare our **BATS** with two latest OOD detection methods KNN [47] and ViM [48]. KNN is a nearest-neighbor-based OOD detection method, which computes the k-th nearest neighbor (KNN) distance between the embedding of test input and the embeddings of the training set to determine if the input is OOD or not. ViM combines the class-agnostic score from feature space and the In-Distribution class-dependent logits to calculate the OOD score. As shown in Tab. 4, our **BATS** outperforms the existing methods by a large margin. KNN explores and demonstrates the efficacy of the non-parametric nearest-neighbor distance for OOD detection, but its performance is

Table 3: Using TFEM to select features' typical set. We use the pre-trained ResNet-50 and ViT-B/16 to detect the OOD examples. The best results are in bold.

| Model | Method | iNaturalist | | SUN | | Places | | Textures | | Average | |
|---|---|---|---|---|---|---|---|---|---|---|---|
| | | FPR95 | AUROC | FPR95 | AUROC | FPR95 | AUROC | FPR95 | AUROC | FPR95 | AUROC |
| ViT | MSP | 16.15 | 96.37 | 56.56 | 85.18 | 59.39 | 84.62 | 50.99 | 84.68 | 45.77 | 87.71 |
| | MSP+**TFEM** | **4.10** | **99.09** | **40.62** | **90.97** | **47.43** | **89.20** | **39.70** | **89.06** | **32.96** | **92.08** |
| | ODIN | 13.90 | 96.88 | 43.91 | 88.89 | 52.19 | 85.90 | 42.36 | 88.35 | 38.09 | 90.01 |
| | ODIN+**TFEM** | **6.45** | **98.78** | **35.44** | **92.39** | **45.36** | **89.34** | 43.07 | **88.61** | **32.58** | **92.28** |
| | Energy | 5.26 | 98.62 | 40.81 | 90.80 | 48.75 | 88.44 | 34.06 | 91.25 | 32.22 | 92.28 |
| | Energy+**TFEM** | **1.48** | **99.68** | **29.19** | **93.84** | **40.12** | **91.22** | **30.44** | **92.17** | **25.31** | **94.23** |
| | GradNorm | 5.14 | 98.35 | 42.06 | 89.26 | 49.21 | 86.63 | 35.57 | 89.27 | 33.00 | 90.88 |
| | GradNorm+**TFEM** | **1.50** | **99.66** | **28.86** | **93.88** | **40.04** | **91.27** | **30.69** | **92.08** | **25.27** | **94.22** |
| ResNet50 | MSP | 51.44 | 88.17 | 72.04 | 79.95 | 74.34 | 78.84 | **54.90** | 78.69 | 63.18 | 81.41 |
| | MSP+**TFEM** | **38.50** | **92.77** | **66.53** | **84.47** | **70.59** | **82.13** | 58.40 | **86.71** | **58.51** | **86.52** |
| | ODIN | 41.07 | 91.32 | 64.63 | 84.71 | 68.36 | 81.95 | 50.55 | 85.77 | 56.15 | 85.94 |
| | ODIN+**TFEM** | **28.40** | **94.67** | **52.34** | **89.47** | **62.13** | **85.14** | **37.27** | **92.35** | **45.04** | **90.41** |
| | Energy | 46.65 | 91.32 | 61.96 | 84.88 | 67.97 | 82.21 | 56.06 | 84.88 | 58.16 | 85.82 |
| | Energy+**TFEM** | **20.29** | **96.24** | **53.98** | **86.85** | **43.37** | **90.90** | **38.24** | **92.22** | **38.97** | **91.55** |
| | GradNorm | 23.73 | 93.97 | 42.81 | 87.26 | 55.62 | 81.85 | 38.15 | 87.73 | 40.08 | 87.70 |
| | GradNorm+**TFEM** | **11.88** | **97.83** | **26.24** | **95.00** | **40.46** | **90.77** | **25.05** | **94.85** | **25.91** | **94.61** |

worse than GradNorm and ReAct. ViM performs well on the OOD dataset Textures, but when using SUN as the OOD dataset, its performance is even worse than the simple baseline MSP.

Table 4: OOD detection performance comparison on ResNet-50 on the ImageNet benchmark. All methods are post hoc and can be directly used for pre-trained models. The best results are in Bold.

| Method | iNaturalist | | SUN | | Places | | Textures | | Average | |
|---|---|---|---|---|---|---|---|---|---|---|
| | FPR95 | AUROC | FPR95 | AUROC | FPR95 | AUROC | FPR95 | AUROC | FPR95 | AUROC |
| MSP [24] | 51.44 | 88.17 | 72.04 | 79.95 | 74.34 | 78.84 | 54.90 | 78.69 | 63.18 | 81.41 |
| ODIN [25] | 41.07 | 91.32 | 64.63 | 84.71 | 68.36 | 81.95 | 50.55 | 85.77 | 56.15 | 85.94 |
| Energy [7] | 46.65 | 91.32 | 61.96 | 84.88 | 67.97 | 82.21 | 56.06 | 84.88 | 58.16 | 85.82 |
| GradNorm [15] | 23.73 | 93.97 | 42.81 | 87.26 | 55.62 | 81.85 | 38.15 | 87.73 | 40.08 | 87.70 |
| ReAct [14] | 17.77 | 96.70 | 25.15 | 94.34 | 34.64 | 91.92 | 51.31 | 88.83 | 32.22 | 92.95 |
| KNN [47] | 59.00 | 86.47 | 68.82 | 80.72 | 76.28 | 75.76 | **11.77** | **97.07** | 53.97 | 85.01 |
| ViM [48] | 77.34 | 86.46 | 90.71 | 73.80 | 89.64 | 72.15 | 16.63 | 96.37 | 68.58 | 82.20 |
| BATS(Ours) | **12.57** | **97.67** | **22.62** | **95.33** | **34.34** | **91.83** | 38.90 | 92.27 | **27.11** | **94.28** |

# Q    Benefits of BATS on calibration

The outputs of a classifier are often interpreted as the predictive confidence that this class was identified. Deep neural networks are often not calibrated which means that the confidence always does not align with the misclassification rate. Expected Calibration Error (ECE) is a metric to measure the calibration of a classifier. For a perfectly calibrated classifier, the ECE value will be zero.

We use the reliability diagram to find out how well the classifier is calibrated in Fig. 11. The model's predictions are divided into bins based on the confidence value of the target class, here, we choose 20 bins. The confidence histogram shows how many test examples are in each bin. Two vertical lines represent the accuracy and average confidence, and the closer these two lines are, the better the model calibration is. BATS can improve the calibration of the pre-trained model and reduce the ECE of the pre-trained model from 3.56% to 2.12%.

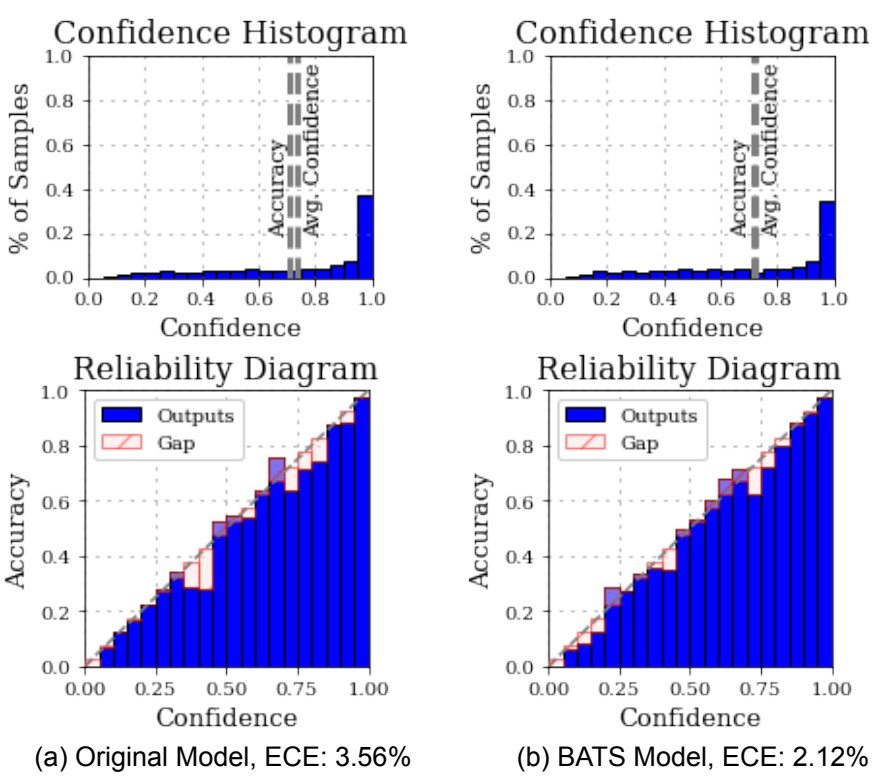

Figure 11: We draw the reliability diagram and the confidence histogram of the pre-trained ResNet-50 (a) and the ResNet-50 with our BATS (b) on ImageNet.