# OpenReview forum: "Boosting Out-of-distribution Detection with Typical Features"
_NeurIPS.cc/2022/Conference — NeurIPS 2022 Accept_

### Official Review · Reviewer_juXE · 2022-06-20

**Rating:** 6
**Confidence:** 5
**Soundness:** 2 fair
**Presentation:** 3 good
**Contribution:** 1 poor

**Summary:**

This paper proposed Batch Normalization Assisted Typical Set Estimation (BATS) for enhancing Out-of-distribution Detection methods. BATS is a truncated activation scheme to bound the output features of the BN unit. The paper demonstrated that BATS could improve several OOD detection methods, such as Energy, Gradnorm, ODIN, when applying it to rectify the features of the penultimate layer. The proposed approach looks simple and effective.

**Questions:**

Besides the concerns mentioned above, I don't understand why BATS can boost OOD detection theoretically.

Section 4.3 shows that BATS reduces the variance of output features and also introduces a bias term. In lines 175 - 177, the authors stated that "Our BATS aids this problem by reducing the variance of the deep features, which contributes to constraining the uncertainty of f and T(x; f) and improving the estimation accuracy of the reject region."

However, why variance reduction boosts OOD detection isn't clear. Does it generally improve classifiers or specifically solve OOD detection problem?

**Limitations:**

The discussion of limitations by authors is adequate. As said in Lines 299 - 300, "The limitation of our method can be that the Batch-Norm layers are required in the model architecture in our approach." I would encourage the authors to continue to improve the method and find a more general formulation, as normalization technique (especially unified formulation) have been extensively studied.

**Strengths And Weaknesses:**

Strengths:
1) The proposed approach is simple to implement
2) The proposed approach is effective on several OOD detection methods with BN in the architecture demonstrated by experiments
3) The paper is well-written and easy to understand.

Weaknesses:
1) My first impression of this paper is that the proposed approach looks like a slight modification of ReAct.
Although BATS outperforms ReAct in experiments, the operation and the mathematical analysis look very similar.
In my understanding, BATS adaptively estimates the parameter c in ReAct.

- The authors explain the motivation in the perspective of typical features, but typicality is not novel in OOD detection, e.g.,

[r1] Detecting Out-of-Distribution Inputs to Deep Generative Models Using Typicality.

[r2] WAIC, but Why? Generative Ensembles for Robust Anomaly Detection.

2) The major limitation of the proposed approach is that it can be applied only on BN (please correct me if I misunderstand.)
This can bring two major concerns:
- As previous research showed that BN increases adversarial vulnerability [r3, r4], whether the proposed approach solved the problem of BN or had the ability of detecting OOD samples is not clear.
- Modern archiectures utilize better normalization methods other than BN, such as LN and GN, and thus interest and impact of the proposed method is greatly limited.

[r3] Batch Normalization Increases Adversarial Vulnerability and Decreases Adversarial Transferability: A Non-Robust Feature Perspective. ICCV 2021.

[r4] Batch Normalization is a Cause of Adversarial Vulnerability.

3) The authors didn't compare with SOTA of OOD detection on large-scale ImageNet, e.g.,

[r5] Out-of-distribution detection with deep nearest neighbors. ICML 2022.

[r6] VIM: Out-of-distribution with virtual-logit matching. CVPR 2022.

---

> ### Author Response · Authors · 2022-08-02
> **Response to Reviewer juXE (Part 1/4)**
>
> Thank you for your thoughtful comments. Below we address the feedback and comments in detail. Please feel free to let us know if you have any further questions about the paper. We will try our best to address your concerns.
>
> ***
> >**Q1:** My first impression of this paper is that the proposed approach looks like a slight modification of ReAct. Although BATS outperforms ReAct in experiments, the operation and the mathematical analysis look very similar. In my understanding, BATS adaptively estimates the parameter c in ReAct.
>
> **A1:** The similarity between our BATS and ReAct is that these methods are used to improve the performance of the existing OOD scores. As follows, we discuss the difference between BATS and ReAct from three aspects.
>
> First, the motivation between our BATS and ReAct is different. ReAct hypothesizes that the mean activation of OOD data has significantly larger variations across units and is biased towards having sharp positive values, while the activation of the ID data is well-behaved with a near-constant mean and standard deviation. Thus, ReAct thinks that the truncation can rectify the activation of the OOD examples and preserve the activation for in-distribution data. However, this hypothesis does not always hold, as shown in Fig. 15 in Appendix N.
> The distribution of the deep features after batch normalization is consistent with the Gaussian distribution. Our BATS hypothesizes that deep models may be hard to model the extreme features but can provide reliable estimations on the typical features. This is because extreme features are exposed to the training process with a low probability. We propose to rectify the features into the typical set and calculate the OOD scores with the typical features.
>
> Second, the mathematical analysis between our BATS and ReAct is different. ReAct theoretically analyze that if the OOD activations are more positively skewed, their operation reduces mean OOD activations more than ID activations.
> We analyze the benefit of BATS from the perspective of the bias-variance trade-off. BATS can reduce the variance of the deep features, which contributes to constraining the uncertainty of the test static $T(x; f)$ and improving the estimation accuracy of the reject region. Our method hopes to estimate the reject region better, and we do not assume whether OOD data is positively skewed.
>
> Third, our method surpasses the ReAct in both the large-scale benchmark (ImageNet) and the small-scale benchmark (CIFAR).
> We have added some discussion in Appendix N in the revised version to make our idea easier to read.
>
> ***
> >**Q2:** The authors explain the motivation in the perspective of typical features, but typicality is not novel in OOD detection[1,2].
>
> **A2:** Thanks for your comments. It's true that the typical set was proposed by Shannon in 1948 [3], which indicates the set whose elements have an information content sufficiently close to that of the expected information. However, the typicality introduced in these density-based methods [1,2] and our proposed BATS are used for different purposes. The typical sets in these density-based OOD detection methods indicate sets of samples whose expected log-likelihood approximates the model's entropy. These density-based OOD detection methods distinguish the OOD examples by estimating whether the examples lie in the typical set of the model.
> In contrast, the typical features in our paper indicate the features that fall into the high-probability regions. We rectify the features into the typical set in order to reduce the variance of the test statistic and improve the estimation accuracy of the reject region. Our method aims to improve the performance of the classification-based OOD detection methods from the perspective of typicality.
>
> Moreover, the density-based methods [1,2] need to train the generative models, which are time-consuming and can hardly adopt in large-scale settings. The performance of the density-based methods can often lag behind the classification-based approaches [4].

---

> ### Author Response · Authors · 2022-08-02
> **Response to Reviewer juXE (Part 2/4)**
>
> >**Q3:** The major limitation of the proposed approach is that it can be applied only on BN (please correct me if I misunderstand.) Modern architectures utilize better normalization methods other than BN, such as LN and GN, and thus interest and impact of the proposed method is greatly limited.
>
> **A3:** In this paper, we provide new insights into classification-based OOD detection from the perspective of typicality and propose to rectify the features into the features' typical set. Regarding how to select the features' typical set, we design a concise and effective approach to select the features' typical set with the assistance of BN layers in our paper. To select the features' typical set without the assistance of the BN layers, we provide another simple way to extend our method to models without BN.
>
> To be specific, we directly use a set of training images to estimate the mean $\mu$ and the standard deviation $\sigma$ of the features (extracted by the penultimate layer of the model) at each dimension. In this experiment, we randomly choose 1500 images from the training dataset of the ImageNet. Then we rectify the features into the interval [$\mu$-$\lambda$*$\sigma$, $\mu$+$\lambda$*$\sigma$] and use these typical features to calculate the OOD scores.
> We name this method as Typical Feature Estimated Method (TFEM) and show the results in the following table. The experiment is performed on the ImageNet benchmark. The pre-trained model is ResNet-50 and ViT. The $\lambda$ is set to 1. Rectifying the features into the typical set with TFEM can greatly improve the performance of the existing OOD detection methods both on the model with BN layers (ResNet-50) and the model without BN layers (ViT).
>
> This experiment demonstrates the effectiveness of the typical features in OOD detection, which is consistent with the analysis in our paper. We believe there exists a method that can estimate the features' typical set better. In this paper, BATS has already established state-of-the-art performance on both the large-scale and small-scale OOD detection benchmarks. We have added this experiment in Appendix O.
>
> |   Model  |    Method    | iNaturalist |        |   SUN  |        | Places |        | Textures |        | Average |        |
> |:--------:|:------------:|:-----------:|:------:|:------:|:------:|:------:|:------:|:--------:|:------:|:-------:|:------:|
> |          |              |    FPR95    |  AUROC |  FPR95 |  AUROC |  FPR95 |  AUROC |   FPR95  |  AUROC |  FPR95  |  AUROC |
> |    ViT   |      MSP     |    18.72    | 96.09  | 56.02  | 85.92  | 59.30  | 84.85  |  51.08   | 84.90  |  46.28  | 87.94  |
> |          |    MSP+TFEM   |    7.12     | 98.34  | 42.62  | 90.36  | 48.71  | 88.64  |  41.29   | 88.48  |  34.94  | 91.46  |
> |          |     ODIN     |    12.72    | 97.15  | 40.04  | 90.40  | 50.46  | 87.10  |  41.08   | 89.30  |  36.08  | 90.99  |
> |          |   ODIN+TFEM   |    8.43     | 98.28  | 33.37  | 93.00  | 44.80  | 89.64  |  39.98   | 89.87  |  31.65  | 92.70  |
> |          |    Energy    |    6.11     | 98.67  | 36.83  | 91.82  | 45.26  | 89.37  |  31.86   | 91.78  |  30.02  | 92.91  |
> |          |  Energy+TFEM  |    3.59     | 99.08  | 30.27  | 93.70  | 41.53  | 90.51  |  31.13   | 92.13  |  26.63  | 93.86  |
> |          |   GradNorm   |    7.66     | 98.43  | 48.36  | 85.28  | 66.38  | 72.85  |  48.79   | 80.76  |  42.80  | 84.33  |
> |          | GradNorm+TFEM |    4.05     | 98.95  | 30.23  | 93.37  | 41.07  | 91.20  |  31.67   | 91.82  |  26.76  | 93.84  |
> | ResNet50 |      MSP     |    51.44    | 88.17  | 72.04  | 79.95  | 74.34  | 78.84  |  54.90   | 78.69  |  63.18  | 81.41  |
> |          |    MSP+TFEM   |    38.50    | 92.77  | 66.53  | 84.47  | 70.59  | 82.13  |  58.40   | 86.71  |  58.51  | 86.52  |
> |          |     ODIN     |    41.07    | 91.32  | 64.63  | 84.71  | 68.36  | 81.95  |  50.55   | 85.77  |  56.15  | 85.94  |
> |          |   ODIN+TFEM   |    28.40    | 94.67  | 52.34  | 89.47  | 62.13  | 85.14  |  37.27   | 92.35  |  45.04  | 90.41  |
> |          |    Energy    |    46.65    | 91.32  | 61.96  | 84.88  | 67.97  | 82.21  |  56.06   | 84.88  |  58.16  | 85.82  |
> |          |  Energy+TFEM  |    20.29    | 96.24  | 53.98  | 86.85  | 43.37  | 90.90  |  38.24   | 92.22  |  38.97  | 91.55  |
> |          |   GradNorm   |    23.73    | 93.97  | 42.81  | 87.26  | 55.62  | 81.85  |  38.15   | 87.73  |  40.08  | 87.70  |
> |          | GradNorm+TFEM |    11.88    | 97.83  | 26.24  | 95.00  | 40.46  | 90.77  |  25.05   | 94.85  |  25.91  | 94.61  |

---

> ### Author Response · Authors · 2022-08-02
> **Response to Reviewer juXE (Part 3/4)**
>
> >**Q4:** As previous research showed that BN increases adversarial vulnerability, whether the proposed approach solved the problem of BN or had the ability of detecting OOD samples is not clear.
>
> **A4:** Empirically, we perform extensive evaluations and establish superior performance on both the large-scale ImageNet OOD detection benchmark and the commonly used CIFAR benchmarks. We focus on the OOD detection task rather than adversarial vulnerability tasks. Actually, our approach can not improve the adversarial robustness but can slightly improve the test accuracy and the robustness of the pre-trained models (as shown in Appendix H). Regarding the influence of the adversarial vulnerability on OOD detection, we found that the normal pre-trained model and the adversarially trained robust models perform similar to each other in detecting the OOD examples and our BATS surpasses the existing methods (we illustrate the performance of different models in Fig.8 in our paper).
>
> ****
> >**Q5:** The authors didn't compare with SOTA of OOD detection [5,6] on large-scale ImageNet.
>
> **A5:** Thanks for your suggestion. The recently published methods KNN [5] and ViM [6] are very interesting works. According to your suggestion, we have added the comparison in Appendix P in the revision.
> KNN is a nearest-neighbor-based OOD detection method, which computes the k-th nearest neighbor (KNN) distance between the embedding of test input and the embeddings of the training set to determine if the input is OOD or not. ViM combines the class-agnostic score from feature space and the In-Distribution class-dependent logits to calculate the OOD score. The following table shows the OOD detection performance of different methods on ResNet-50 on the ImageNet benchmark. Our BATS outperforms the existing methods by a large margin. KNN explores and demonstrates the efficacy of the non-parametric nearest-neighbor distance for OOD detection, but its performance is worse than GradNorm and ReAct. ViM performs well on the OOD dataset Textures, but when using SUN as the OOD dataset, its performance is even worse than the simple baseline MSP.
>
> |  Method  | iNaturalist |        |   SUN  |        | Places |        | Textures |        | Average |        |
> |:--------:|:-----------:|:------:|:------:|:------:|:------:|:------:|:--------:|:------:|:-------:|:------:|
> |          |    FPR95    |  AUROC |  FPR95 |  AUROC |  FPR95 |  AUROC |   FPR95  |  AUROC |  FPR95  |  AUROC |
> |    MSP   |    51.44    | 88.17  | 72.04  | 79.95  | 74.34  | 78.84  |  54.90   | 78.69  |  63.18  | 81.41  |
> |   ODIN   |    41.07    | 91.32  | 64.63  | 84.71  | 68.36  | 81.95  |  50.55   | 85.77  |  56.15  | 85.94  |
> |  Energy  |    46.65    | 91.32  | 61.96  | 84.88  | 67.97  | 82.21  |  56.06   | 84.88  |  58.16  | 85.82  |
> | GradNorm |    23.73    | 93.97  | 42.81  | 87.26  | 55.62  | 81.85  |  38.15   | 87.73  |  40.08  | 87.70  |
> |   ReAct  |    17.77    | 96.70  | 25.15  | 94.34  | 34.64  | 91.92  |  51.31   | 88.83  |  32.22  | 92.95  |
> |    KNN   |    59.00    | 86.47  | 68.82  | 80.72  | 76.28  | 75.76  |  11.77   | 97.07  |  53.97  | 85.01  |
> |    VIM   |    77.34    | 86.46  | 90.71  | 73.80  | 89.64  | 72.15  |  16.63   | 96.37  |  68.58  | 82.20  |
> |   Ours   |    12.57    | 97.67  | 22.62  | 95.33  | 34.34  | 91.83  |  38.90   | 92.27  |  27.11  | 94.28  |

---

> ### Author Response · Authors · 2022-08-02
> **Response to Reviewer juXE (Part 4/4)**
>
> >**Q6:** Besides the concerns mentioned above, I don't understand why BATS can boost OOD detection theoretically. Why variance reduction boosts OOD detection isn't clear. Does it generally improve classifiers or specifically solve OOD detection problem? }
>
> **A6:** Thanks for your comments.
> Intuitively, as shown in Fig. 1, there exists an overlap in the distribution of the scores for ID and OOD examples. Smaller overlap indicates better OOD detection performance.
> BATS constrains the variance of the distribution of the OOD score, which reduces the overlap between the ID and OOD examples and improves the separability between the ID and OOD examples. Moreover, we added an illustration (Fig. 10) in Appendix I to show the influence of our BATS on different OOD detection methods. These detection methods hope to assign higher scores for the ID examples and lower scores for the OOD examples. BATS can reduce the variance of the scores and the overlap between the distribution of ID and OOD examples, which benefits OOD detection.
>
> Mathematically, as we analyze in the preliminaries and Appendix A in our paper, OOD detection is a single-sample hypothesis testing problem.
> Let $X$ be the input space. Suppose that the in-distribution data $D_{in}$ is drawn from a distribution $P_{0}$ defined over $X.$
> Given a test input $x \in X$, the problem of out-of-distribution  detection can be formulated as a single-sample hypothesis testing task:
>
> $H_0: x \sim P_0, \quad \text{vs.} \quad H_1: x \nsim P_0. $
>
> Here the null hypothesis $H_0$ implies that the test input $x$ is an in-distribution sample.
> The goal of OOD detection here is to design criteria based on $D_{in}$ to determine whether $H_0$ should be rejected. OOD detection tasks need to determine a reject region $R$ such that for any test input $x \in X$, the null hypothesis is rejected if $x \in R.$
> Generally, the reject region $R$ is formulated by a test statistic and a threshold.
> Let $f$ be a  model pre-trained from $D_{in}$, which is used to predict the class label of an input sample. One can use the model $f$ or a part of $f$ (e.g., feature extractor) to construct a test statistic (also named as OOD score in OOD detection literature) $T(x; f)$, where $x$ is the test input. Then the reject region can be written as $R = {x: T(x;f) \leq \gamma}$, where $\gamma$ is the threshold.
> Because the in-distribution $P_0$ is unknown, the reject region is determined by the empirical distribution of the test statistic $T(x; f)$ over the ID data. If the test statistic $T(x; f)$ over the ID data has a large variance and contains many unusual values, the reject region may be underestimated. By reducing the variance, BATS constrains the uncertainty of the test statistic and can improve the estimation accuracy of the reject region.
>
> ***
> **References**
>
> [1] Nalisnick, et al. "Detecting Out-of-Distribution Inputs to Deep Generative Models Using a Test for Typicality." (2019).
>
> [2] Choi H, et al. "WAIC, but Why? Generative Ensembles for Robust Anomaly Detection." (2018).
>
> [3] Shannon C E. "A mathematical theory of communication". (1948).
>
> [4] Yang, Jingkang, et al. "Generalized out-of-distribution detection: A survey." (2021).
>
>
> [5] Sun, Yiyou, et al. "Out-of-distribution Detection with Deep Nearest Neighbors." (2022).
>
> [6] Wang, Haoqi, et al. "ViM: Out-Of-Distribution with Virtual-logit Matching." (2022).
>
> [7] Haroush, Matan, et al. "A Statistical Framework for Efficient Out of Distribution Detection in Deep Neural Networks." (2021).

---

> ### Author Response · Authors · 2022-08-07
> **Further Discussion with Reviewer juXE**
>
> We thank you for the precious review time and valuable comments. We have provided corresponding responses and results, which we hope to address your concerns. We hope to further discuss with you whether or not your concerns have been addressed appropriately. Please let us know if you have additional questions or ideas for improvement.

---

> > ### Comment · Reviewer_juXE · 2022-08-09
> > **Response to Authors**
> >
> > I appreciate that the authors took the time to carry out the analysis suggested by all reviewers. After careful reading, I have decided to increase my score as a result. This is a strong paper with valuable contributions and insights.

---

> > > ### Author Response · Authors · 2022-08-09
> > > **Thank You**
> > >
> > > Thanks again for your precious review time and valuable comments. Best regards!

---

### Official Review · Reviewer_NEkg · 2022-07-10

**Rating:** 7
**Confidence:** 3
**Soundness:** 3 good
**Presentation:** 3 good
**Contribution:** 3 good

**Summary:**

The paper proposes a replacement of the batch normalization layer (BATS) to rectify deep model features into its typical set to improve OOD detection performance. The authors provide theorectical analysis and ablation studies to look deeper into BATS, and achieves state-of-the-art performance using it as a play-and-plug module.

**Questions:**

1.Uncertainty measurement is important in OOD cases. Although the authors used FPR95, AUROC and test accuracy (in appendix) to show the effectiveness of BATS, it would be interesting to analyze the results of the pretrained models from the perspective of uncertainty quantification, such as comparing Brier score or expected calibration error (ECE)[1].

2.What are the choices for the reject region threshold $\gamma$, is it always the best value or a fixed value?

[1] Guo C , Pleiss G , Yu S , et al. On Calibration of Modern Neural Networks, 2017.

**Limitations:**

The authors have properly addressed the limitations and potential negative societal impacts.

**Strengths And Weaknesses:**

**Strengths**

Originality: The idea is new and intriguing. The authors propose a novel insight for OOD detection from the perspective of feature typicality, and divides deep features into typical features and extreme features. They also propose a novel replacement for Batch Normalization, which can be integrated into existing model structrues and OOD scores.

Quality: From the perspective of typicality, and under the assumption that extreme features are harmful for model training, the authors provide thourough ablation studies on the proposed BATS approach, and theoretically prove its bias-variance trade-off. The proof is in detail and experiments are discussed well.

Clarity: The paper is well-written and easy to follow.

Significance: The paper provides a novel perspective into OOD detection, and introduces a simple yet effective method for post-hoc detection. The method is evaluated on both small and large real-world datasets.

**Weaknesses**

I have not identified major weaknesses of this paper, while I do have some minor concerns that are listed in the “Questions” part.

---

> ### Author Response · Authors · 2022-08-02
> **Response to Reviewer NEkg**
>
> Thank you for your positive assessment and helpful feedback. We appreciate the time and attention you spent on reviewing our paper.
>
> ***
> >**Q1:** Uncertainty measurement is important in OOD cases. It would be interesting to analyze the results of the pretrained models from the perspective of uncertainty quantification, such as comparing Brier score or expected calibration error (ECE).
>
> **A1:** Thanks for your suggestion. We found that BATS can improve the calibration of the pre-trained model and reduce the expected calibration error (ECE) of the pre-trained ResNet-50 from 3.56% to 2.12%. We have added the reliability diagram and the confidence histogram of the pre-trained ResNet-50 and the ResNet-50 with our BATS on ImageNet in Appendix Q in the revision.
>
> ***
> >**Q2:** What are the choices for the reject region threshold $\gamma$, is it always the best value or a fixed value?
>
> **A2:** Thanks for your comments. Following the standard settings in the existing works, the reject region threshold is a fixed value, which can correctly identify 95% in-distribution examples as in-distribution examples (the true positive rate of in-distribution (positive) examples is 95%).

---

> > ### Comment · Reviewer_NEkg · 2022-08-07
> > **Further comment**
> >
> > The authors have addressed my concerns and I appreciate the efforts the authors made to refine the paper. Though I would like this paper to be accepted, I am willing to hear about the other reviewers' further opinions, especially reviewer juXE, whose score is divergent.

---

> > > ### Author Response · Authors · 2022-08-07
> > > **Thank You**
> > >
> > > Thank you very much for your valuable review. We are grateful that you appreciate our paper.

---

### Official Review · Reviewer_3fd9 · 2022-07-11

**Rating:** 7
**Confidence:** 4
**Soundness:** 3 good
**Presentation:** 2 fair
**Contribution:** 3 good

**Summary:**

The authors propose an OoD method that does not require the retraining of the model, and which relies on replacing the last BN layer by their proposed TrBN at inference time. The main concept of the new layer is that it clamps the most extreme features to be within their variance. By correcting those features (compressing the extreme features), the outputs of the model become less susceptible to OoD miss-classifications. The authors provide comparable or better results to existing state-of-the-art methods under different OoD scenarios.

**Questions:**

Could you provide a bit more explanation on the differences with ReAct, that truncates the values of the activations? I assume by the results that the activations targeted by both ReAct and BATS are not the same, but is there any insight on that? Are the abnormal activations of ReAct closely correlated with the extreme features of BATS?

Most reported results are basically an upgrade on top of Energy [5], which can be applied to other methods (as shown in Fig. 3, and Appendix I). Therefore, wouldn't it make more sense to report it in most tables/figures as Energy+BATS? Considering the huge gap between the Energy and the Energy+BATS results, why use that combination and not another one? Even with the added cost of GradNorm, the results are much better. Also, that would raise the question to why would BATS work better in some methods than others.

Minor comments:
- check for typos and revise manuscript (e.g. line 9 play-and-plug --> plug-and-play, line 98 energe --> energy)
- Table 2, CIFAR100 WRN, ODIN, that should be in bold (check for other cases)
- Figure 5 is not mentioned in the text.

**Limitations:**

It is correctly stated in the limitations that this method is proposed as a post-hoc method, however, it has the limitation of the model a BN layer before the last FC layer.

There does not seem to be any potential negative societal impact generated from this work.

**Strengths And Weaknesses:**

*Strentghs*

The idea is of interest to the community and the method is easy to implement.

The theoretical analysis seems sound and well based.

The analysis of the trade-off from lambda and the bias introduced by the proposed TrBN looks correct.

The ablation study provides nice insight and answers a question I was already thinking of while reading the method section.

*Weaknesses*

The comparison with the related work is brief and there is little discussion of the similarities and differences. Methods such as ReAct seem to be close to the proposed approach, since they target "anomalous features".

The presentation could be improved. It requires a revision to correct some typos and grammatical errors. Some images and their texts could be more readable (e.g. Figs. 1-2).

The best results of the proposed method are in the supplementary, which is a bit odd. I would expect those to be discussed in the main paper, and also provide some insight on why the proposed method might work better in some of those methods. With the results shown in the main paper, the improvements in some of the scenarios are quite marginal in relation to other methods.

---

> ### Author Response · Authors · 2022-08-02
> **Response to Reviewer 3fd9**
>
> We sincerely appreciate your appreciation of our paper and the positive comments. We address specific questions below.
>
> ***
>
> >**Q1:** The comparison with the related work is brief and there is little discussion of the similarities and differences. Methods such as ReAct seem to be close to the proposed approach since they target "anomalous features".
>
> **A1:** Thanks for your suggestion. Due to page limitations, we place some related literature in Appendix G. According to your suggestion, we have added some discussion on the difference between BATS and ReAct in Appendix N.
>
> First, the motivation between our BATS and ReAct is different. ReAct hypothesizes that the mean activation of OOD data has significantly larger variations across units and is biased towards having sharp positive values, while the activation of the ID data is well-behaved with a near-constant mean and standard deviation. Thus, ReAct thinks that the truncation can rectify the activation of the OOD examples and preserve the activation for in-distribution data.
> However, this hypothesis does not always hold, as shown in Fig. 15 in Appendix N. The distribution of the deep features after batch normalization is consistent with the Gaussian distribution. Our BATS hypothesizes that deep models may be hard to model the extreme features but can provide reliable estimations on the typical features. This is because extreme features are exposed to the training process with a low probability. We propose to rectify the features into the typical set and calculate the OOD scores with the typical features.
>
> Second, the mathematical analysis between our BATS and ReAct is different. ReAct theoretically analyze that if the OOD activations are more positively skewed, their operation reduces mean OOD activations more than ID activations.
> We analyze the benefit of BATS from the perspective of the bias-variance trade-off. BATS can reduce the variance of the deep features, which contributes to constraining the uncertainty of the test static $T(x; f)$ and improving the estimation accuracy of the reject region. Our method hopes to estimate the reject region better, and we do not assume whether OOD data is positively skewed.
>
> Third, our method surpasses the ReAct in both the large-scale benchmark (ImageNet) and the small-scale benchmark (CIFAR).
>
> ****
> >**Q2:** The presentation could be improved. It requires a revision to correct some typos and grammatical errors. Some images and their texts could be more readable.
>
> **A2:** Thanks for your suggestion. We have carefully checked the typos and improved the writing in the revised manuscript.
>
> ****
> >**Q3:** The best results of the proposed method are in the supplementary, which is a bit odd. I would expect those to be discussed in the main paper, and also provide some insight on why the proposed method might work better in some of those methods.
>
> **A3:** In this paper, we hope to provide new insights into OOD detection from the perspective of typicality and show that rectifying the features into the typical set can greatly improve the performance of the OOD scores. Our proposed method is compatible with many test statistics (OOD scores).
> Experimentally, we mainly show that applying our method to the normally used Energy score can achieve state-of-the-art performance. Moreover, Tab. 5 in our paper shows that our method can further improve the performance of other OOD scores, specifically that higher performance can be obtained using more advanced OOD scores (GradNorm).
>
> We have added an illustration (Fig.10) in Appendix I in the revision.
> GradNorm itself performs better than the simple baseline method MSP score, which can assign higher scores to the ID examples and lower scores to the OOD examples.
> Applying BATS to the existing OOD detection methods can reduce the variance of the scores and reduce the overlap between the ID examples and the OOD examples. We think combining our method with a better OOD score can achieve better performance.

---

> > ### Comment · Reviewer_3fd9 · 2022-08-09
> > **Response**
> >
> > Thanks to the authors for their thorough answers to my comments and to all other reviewers. I think the responses and the modifications to the manuscript cover my questions appropriately and I found some of the answers to other reviewers equally interesting. I therefore consider this manuscript should be accepted.

---

> > > ### Author Response · Authors · 2022-08-09
> > > **Thank You**
> > >
> > > Thanks again for your appreciation of our paper and the valuable comments. Best regards.

---

### Official Review · Reviewer_bMZr · 2022-07-12

**Rating:** 7
**Confidence:** 3
**Soundness:** 3 good
**Presentation:** 3 good
**Contribution:** 3 good

**Summary:**

The paper presents a post hoc method for OOD detection in classification models. The key to the proposed approach is to determine the typical feature set from the data and the trained neural network, and use it to compute an existing OOD detection score, such as the energy score. The typical set is computed by truncating the output from a batch normalization error. The truncation is controlled by a hyperparameter that determines the bias-variance trade-off for the method. The proposed BATS method outperforms baseline methods consistently across a variety of experimental settings.

**Questions:**

In addition to addressing the above listed weaknesses, I have the following questions for the authors.

1. How well do you believe that OOD detection can work post hoc without retraining the model? How much do poorly calibrated softmax uncertainties hinder post hoc method effectiveness?
2. Do you have a hypothesis for why BATS performance is lower than baseline methods on the Tiny-Imagenet OOD dataset in Table 2?
3. In the limitations, it is mentioned that 'some other information in the model' may be conducive to selecting the feature's typical set. Do you have a hypothesis for what this information might be?

**Limitations:**

The authors have done a good job in listing limitations of the BATS method. However, the addition of some potential negative societal implications would be helpful. For example, by truncating the features, certain biases in the data learned by the pre-trained model may be amplified. Furthermore, the process of truncation will inherently cause some information loss which may be crucial to model performance during deployment (building on the hyperparameter tuning discussion in the paper).

**Strengths And Weaknesses:**

Strengths:
* I really enjoyed reading this paper. It is well written and the ideas are easy to follow.
* The problem is well-motivated and the literature review does a good job at contextualizing the paper in prior work.
* The proposed approach is simple, yet appears to be highly effective. The theoretical analysis further provides intuition for the BATS method.
* Overall, the empirical evaluation of the method is extensive and convincing. The analysis and discussion is thoughtfully constructed, and the effects of the hyperparameter thoroughly ablated.
* The figures are informative and effectively illustrate the benefits of the proposed approach.
* The analysis in the appendix was extensive and interesting, providing further support for the claims made in the main paper.

Weaknesses:
* In lines 3-4, the statement is not necessarily true as there are also methods that focus on better distribution calibration during training (some of these ideas are mentioned later in the paper as well). I would recommend softening the message of this sentence to be more precise.
* Language like 'Obviously' (line 47) and 'It is easy to see' (line 168) should generally be avoided in academic writing.
* At the end of Sec. 2, it would be helpful to have a one or two sentence discussion to contextualize the proposed BATS method in the described related work.
* Variations on the phrase 'We propose to rectify the features into the feature's typical set and then use these typical features to calculate the OOD score.' are repeated frequently throughout the paper.
* The $\lambda$ hyperparameter was not introduced in line 142 when it was first used.
* The decreased BATS performance on Tiny-Imagenet OOD detection in Table 2 is not mentioned or discussed.

The paper needs to be proofread for typos. The following is a non-exhaustive list of the typos I found:

1. Line 2: 'which raises the attention on out-of-distribution (OOD) detection' is awkward phrasing.
2. Lines 74-75: 'large sufficiently' should read 'sufficiently large'.
3. Line 98: 'energe score' should read 'energy score'.
4. Line 109: 'is provable aligned' should read 'is provably aligned'.
5. Line 130: 'common-used layer' should read 'commonly used layer'.
6. Footnote 1: I did not grammatically understand the phrase: 'the pre-training outputs moving average estimators during iterations'. Maybe something like: 'The pre-trained model outputs moving average estimators at each iteration.'?
7. Line 184: 'a two-side rectified normal distribution' should read 'a two-sided rectified normal distribution'.
8. There should be a space between the abbreviation and the number in references (i.e., Fig. X, Table Y, Sec. Z).
9. Line 197: 'Fig.2 illustrate' should read 'Fig. 2 illustrates'.
10. Line 220: 'verse vice' should read 'vice versa'.
10. Line 225: 'Recent researches propose' should read 'Recent literature/work proposes'.
11. Line 235: 'models are standard pre-trained' is grammatically awkward. Maybe something like: 'models are pre-trained in a standard manner'?
12. Line 238: 'In specific,' should read 'Specifically,'.
13. Line 246: 'which cost more' should read 'which costs more'.
14. Line 252: 'The start learning rate' should read 'The starting learning rate'.
15. Line 284: 'Our BATS can reduce the variance that benefit the OOD detection but also introduce a bias.' should read 'Our proposed BATS method can reduce variance, which benefits OOD detection, but can also introduce a bias.'.
16. Line 285: 'Energy Score (The horizontal lines).' should read 'Energy Score (the horizontal lines)'.
17. Fig. 5: x-axis labels are missing. This is also the case for some figures in the appendix.
18. Sec. 6: 'Limitation and societal impact' should read 'Limitations and societal impact'.
19. Sec. 6: batch normalization is referred to in three different ways in the last paragraph (Batch Normalization, Batch-Norm, BN).
20. Line 693: 'our method surpass' should read 'our method surpasses'.
21. The references should be proofread (e.g., to ensure the year is not entered twice in a citation, the conference venue is listed instead of ArXiv when available, etc.).

---

> ### Author Response · Authors · 2022-08-02
> **Response to Reviewer bMZr (Part 1/2)**
>
> Thank you for the positive feedback and helpful suggestions. We answer your questions point-by-point as follows.
>
> ****
> >**Q1:** In Lines 3-4, the statement is not necessarily true as there are also methods that focus on better distribution calibration during training (some of these ideas are mentioned later in the paper). I would recommend softening the message of this sentence to be more precise.
>
> **A1:** Thank you for your suggestion. In the revision, we have rewritten the sentence "Existing OOD detection methods primarily work for designing OOD scores or introducing diverse outlier examples to retrain the model. We delve into the obstacle factors in OOD detection from the perspective of typicality and regard the feature's high-probability region of the deep model as the feature's typical set." as "Different from most previous OOD detection methods that focus on designing OOD scores or introducing diverse outlier examples to retrain the model, we delve into the obstacle factors in OOD detection from the perspective of typicality and regard the feature's high-probability region of the deep model as the feature's typical set."
>
> ****
> >**Q2:** Language like 'Obviously' (line 47) and 'It is easy to see' (line 168) should generally be avoided in academic writing.
>
> **A2:** Thank you for your suggestion. We have removed these words in the revised version of this paper.
>
> ****
> >**Q3:** At the end of Sec. 2, it would be helpful to have a one or two sentence discussion to contextualize the proposed BATS method in the described related work.
>
> **A3:** We have added the sentence "Different from these methods, our BATS proposes to calculate the OOD scores with the typical features, which benefits the estimation of the reject region and can improve the detection performance." following the related work in Sec.2.
>
> ****
> >**Q4:** Variations on the phrase 'We propose to rectify the features into the feature's typical set and then use these typical features to calculate the OOD score.' are repeated frequently throughout the paper.
>
> **A4:** Thanks for your suggestion. We have removed some unessential phrases in our revised version.
>
> ****
> >**Q5:** The hyperparameter $\lambda$ was not introduced in line 142 when it was first used.
>
> **A5:** Thanks for the helpful comment. The hyperparameter $\lambda$ controls the range of the interval. Larger $\lambda$ indicates higher probability that features fall in the interval  [$\mu$-$\lambda$*$\sigma$,$\mu$+$\lambda$*$\sigma$]. We have added this explanation in the revised version.
>
> ****
> >**Q6:** The decreased BATS performance on Tiny-Imagenet OOD detection in Table 2 is not mentioned or discussed.
>
> **A6:** Thanks for pointing this out. We agree that the decreased performance of our BATS on Tiny-Imagenet OOD detection is interesting and needs some discussion.
>  We hypothesize that this performance degradation is due to the bias introduced by BATS. By truncating the features, BATS can reduce the variance of the in-distribution examples, which benefits the estimation of the reject region but inherently cause some information loss which may reduce the performance of the pre-trained models.
>
> To validate our hypothesis, we tune the bias-variance trade-off by the hyperparameter $\lambda$. As shown in Fig. 14, BATS can indeed reduce the variance of the OOD scores. Choosing a proper $\lambda$, BATS can reduce the overlap between the ID and OOD examples and reduce the FPR95, while a small $\lambda$ hinders the performance of OOD detection. For example, using larger $\lambda=8$, BATS can achieve better FPR95 performance 15.10% on detecting Tiny-Imagenet using ResNet-18, which is 2.65% better than $\lambda=3$ in our Tab. 2. For the practicability of our method, we set the same hyperparameter to test different OOD datasets, without adjusting for specific OOD datasets. We have added this discussion in Appendix M in the revised version.
>
> ****
> >**Q7:** The paper needs to be proofread for typos. The following is a non-exhaustive list of the typos I found.
>
> **A7:** Thanks very much for pointing out the typos in our paper. We really appreciate you for your carefulness and conscientiousness. We have carefully checked the typos and improved the writing in the revised manuscript.
>
> ****
> >**Q8:** How well do you believe that OOD detection can work post hoc without retraining the model?
>
> **A8:** Post-hoc detection methods can be easily adopted in real-world scenarios and large-scale settings because these methods do not need retraining the model. Fig. 4 and Fig. 7 illustrate the t-SNE plots for the features of ID examples and OOD examples, which shows that deep models can extract separable features for the ID and OOD examples.
> Based on this phenomenon, we believe that the deep models know which examples are OOD samples. With the development of interpretability for deep models, a good post-hoc detection method may be designed and achieves excellent detection performance without retraining the model.

---

> ### Author Response · Authors · 2022-08-02
> **Response to Reviewer bMZr (Part 2/2)**
>
> >**Q9:** How much do poorly calibrated softmax uncertainties hinder post hoc method effectiveness?
>
> **A9:** We provide an experiment for the influence of the calibration of the softmax uncertainty on the OOD detection. Using the temperature scaling method [3], we get one overconfident ResNet-50 (Expected Calibration Error (ECE): 11.59%) and one unconfident ResNet-50 (ECE: 14.52%). The ECE of the original ResNet-50 is 3.56%. In the following table, we compare the performance of different OOD detection methods with different models. "Original" means using the original ResNet-50, "Unconfident" means using the unconfident ResNet-50 and "Overconfident" means using the overconfident ResNet-50. The unconfident ResNet-50 performs better than the original ResNet-50 and overconfident ResNet-50 when using the MSP score, while the unconfident ResNet-50 performs much worse when using the Energy score. The influence of the calibration of the softmax uncertainty on the post hoc method may not be monotonous.
>
> |  Method  |     Model     | iNaturalist |        |   SUN  |        | Places |        | Textures |        | Average |        |
> |:--------:|:-------------:|:-----------:|:------:|:------:|:------:|:------:|:------:|:--------:|:------:|:-------:|:------:|
> |          |               |    FPR95    |  AUROC |  FPR95 |  AUROC |  FPR95 |  AUROC |   FPR95  |  AUROC |  FPR95  |  AUROC |
> |    MSP   |    Original   |    51.44    | 88.17  | 72.04  | 79.95  | 74.34  | 78.84  |  54.90   | 78.69  |  63.18  | 81.41  |
> |          |  Unconfident  |    43.14    | 91.88  | 66.01  | 83.70  | 69.47  | 81.84  |  61.12   | 83.39  |  59.94  | 85.20  |
> |          | Overconfident |    67.23    | 82.56  | 79.28  | 76.50  | 80.73  | 75.67  |  78.87   | 74.28  |  76.53  | 77.25  |
> |  Energy  |    Original   |    46.65    | 91.32  | 61.96  | 84.88  | 67.97  | 82.21  |  56.06   | 84.88  |  58.16  | 85.82  |
> |          |  Unconfident  |    60.88    | 88.03  | 63.71  | 84.59  | 71.49  | 81.28  |  57.04   | 84.50  |  63.28  | 84.60  |
> |          | Overconfident |    44.98    | 91.65  | 62.31  | 84.77  | 67.38  | 82.29  |  56.54   | 84.66  |  57.80  | 85.84  |
> | GradNorm |    Original   |    23.73    | 93.97  | 42.81  | 87.26  | 55.62  | 81.85  |  38.15   | 87.73  |  40.08  | 87.70  |
> |          |  Unconfident  |    30.00    | 93.78  | 45.22  | 89.02  | 58.40  | 84.62  |  40.85   | 88.56  |  43.62  | 89.00  |
> |          | Overconfident |    33.51    | 91.33  | 50.21  | 84.28  | 63.38  | 77.81  |  45.11   | 85.16  |  48.05  | 84.65  |
> |   BATS   |    Original   |    12.57    | 97.67  | 22.62  | 95.33  | 34.34  | 91.83  |  38.90   | 92.27  |  27.11  | 94.28  |
> |          |  Unconfident  |    13.32    | 97.24  | 20.90  | 95.71  | 33.62  | 92.02  |  36.52   | 92.00  |  26.09  | 94.24  |
> |          | Overconfident |    16.57    | 97.07  | 36.68  | 93.01  | 46.81  | 89.81  |  38.76   | 92.15  |  34.71  | 93.01  |
>
> ****
> >**Q10:** Do you have a hypothesis for why BATS performance is lower than baseline methods on the Tiny-Imagenet OOD dataset in Table 2?}
>
> **A10:** Thanks for your comments. We answer this question in A6 above. We have added some discussion in Appendix M in the revised version.
>
> ****
> >**Q11:** In the limitations, it is mentioned that 'some other information in the model' may be conducive to selecting the feature's typical set. Do you have a hypothesis for what this information might be?}
>
> **A11:** First, we think the gradient of the model may be conducive to selecting the feature's typical set because the gradients of the model contain some information about the training data [1]. Second, we think the other parameters of the deep models can also be helpful. For example, the centers of different classes are encoded in the fully connected layer [2], which may be helpful in selecting the typical features. Third, a set of training images can be helpful in calculating the mean and the standard deviation of the features. We have added experiments in Appendix O to show that selecting the features' typical set without the assistance of BN can also greatly improve the performance of the existing OOD detection methods.
>
> ***
>
> **References**
>
> [1] Zhu L, Liu Z, Han S. "Deep leakage from gradients." Advances in neural information processing systems, 2019, 32.
>
> [2] Qian, Qi, et al. "Softtriple loss: Deep metric learning without triplet sampling." Proceedings of the IEEE/CVF International Conference on Computer Vision. 2019.
>
> [3] Guo, Chuan, et al. "On calibration of modern neural networks." International conference on machine learning. 2017.

---

> > ### Comment · Reviewer_bMZr · 2022-08-03
> > **Response to Authors**
> >
> > Thank you for the thorough responses to my questions/concerns and the additional experiments, particularly the discussion on the limitations in Table 2/Appendix M and that on the calibration of the softmax uncertainties! My only remaining concern is in the limitations section. The sentence "We anticipate no negative consequences of our work." should be replaced in my opinion. In my original review, I recommended some potential additions to this section:
> >
> > > The authors have done a good job in listing limitations of the BATS method. However, the addition of some potential negative societal implications would be helpful. For example, by truncating the features, certain biases in the data learned by the pre-trained model may be amplified. Furthermore, the process of truncation will inherently cause some information loss which may be crucial to model performance during deployment (building on the hyperparameter tuning discussion in the paper).

---

> > > ### Author Response · Authors · 2022-08-04
> > > **Response to Reviewer bMZr**
> > >
> > > Thank you for your helpful suggestion. In the revision, we have replaced the sentence "We anticipate no negative consequences of our work." with "Regarding potential negative impact, truncating the features into the typical set can improve the OOD detection but will introduce a bias and inherently cause some information loss which may be important for the model in the real-world scenario."

---

> > > > ### Comment · Reviewer_bMZr · 2022-08-09
> > > > **Minor Edit**
> > > >
> > > > A potential minor edit to the sentence: "Although truncating features into a typical set can improve OOD detection, a potential negative impact of the proposed process is that it inherently introduces a bias and causes some information loss which may be important to the model in real-world scenarios.".

---

> > > > > ### Author Response · Authors · 2022-08-09
> > > > > **Thank You**
> > > > >
> > > > > We really appreciate your valuable comments. Thank you again for helping us improve the manuscript. We have updated the revision.
> > > > > Best wishes!

---

### Author Response · Authors · 2022-08-02
**Update Manuscript**

We thank all of the reviewers for helping us improve the paper. We uploaded a revised version of our manuscript and marked the major changes in blue. In short,

1. We have carefully checked the typos and improved the writing in the revised manuscript.

2. We have added some discussion for the results in Tab.2 in Appendix M.

3. We have added some discussion on the differences between BATS and ReAct in Appendix N.

4. To select the features' typical set without the assistance of the BN layers, we provide another simple way to extend our method to models without BN in Appendix O.

5. We have added an experiment to show that BATS can surpass the other two latest OOD detection methods (KNN Score (ICML2022) and ViM (CVPR2022)) in Appendix P.

6. We find that our BATS can also improve the calibration of the model and show the results in Appendix Q.

Thank you all again for your valuable and insightful suggestions. Please let us know if you have additional questions or ideas for improvement.

Kind regards, Authors

---

### Meta-Review · Area_Chair_EkXz · 2022-08-26

**Recommendation:** Accept
**Confidence:** Certain

**Metareview:**

This paper received unanimous recommendations of acceptance. Concerns were expressed regarding the similarity between the proposed method and ReAct, but the concerns were addressed by the authors. The AC agrees with the reviewer regarding the contribution of this paper and recommends acceptance.

**Award:**

No

---

### Decision · Program_Chairs · 2022-09-14

Accept